# Nuisances via Negativa:
# Adjusting for Spurious Correlations via Data Augmentation

**Aahlad Puli**[1,*]**, Nitish Joshi**[1]**, Yoav Wald**[2]**, He He**[1,2]**, Rajesh Ranganath**[1,2,3]
*Department of Computer Science*[1]*, Center for Data Science*[2]*, Department of Population Health*[3]
*New York University*

Reviewed on OpenReview: *https://openreview.net/forum?id=RIFJsSzwKY*

## Abstract

In prediction tasks, there exist features that are related to the label in the same way across different settings for that task; these are semantic features or semantics. Features with varying relationships to the label are nuisances. For example, in detecting cows from natural images, the shape of the head is semantic but because images of cows often have grass backgrounds but not always, the background is a nuisance. Models that exploit nuisance-label relationships face performance degradation when these relationships change. Building models robust to such changes requires additional knowledge beyond samples of the features and labels. For example, existing work uses annotations of nuisances or assumes ERM-trained models depend on nuisances. Approaches to integrate new kinds of additional knowledge enlarge the settings where robust models can be built. We develop an approach to use knowledge about the semantics by corrupting them in data, and then using the corrupted data to produce models which identify correlations between nuisances and the label. Once these correlations are identified, they can be used to adjust for where nuisances drive predictions. We study semantic corruptions in powering different spurious-correlation avoiding methods on multiple out-of-distribution (OOD) tasks like classifying waterbirds, natural language inference (NLI), and detecting cardiomegaly in chest X-rays.

## 1 Introduction

Relationships between the label and the covariates can change across data collected at different places and times. For example, in classifying animals, data collected in natural habitats have cows appear more often on grasslands, while penguins appear more often on backgrounds of snow; these animal-background relationships do not hold outside natural habitats (Beery et al., 2018; Arjovsky et al., 2019). Some features, like an animal's shape, are predictive of the label across all settings for a task; these are *semantic features*, or *semantics* in short. Other features with varying relationships with the label, like the background, are nuisances. Even with semantics present, models trained via empirical risk minimization (ERM) can predict using nuisances and thus fail to generalize (Geirhos et al., 2020). Models that rely only on the semantic features perform well even when the nuisance-label relationship changes, unlike models that rely on nuisances.

Building models that generalize under changing nuisance-label relationships requires additional knowledge, beyond a dataset of features and labels sampled from the training distribution. For example, many works assume knowledge of the nuisance. In the animal-background example, this would correspond to a feature that specifies the image background, which we may use when specifying our learning algorithm. (Mahabadi et al., 2019; Makar et al., 2022; Veitch et al., 2021; Puli et al., 2022); another common type of assumption is access to multiple datasets over which the nuisance-label correlation varies (Peters et al., 2016; Arjovsky et al., 2019; Wald et al., 2021), and some other forms of knowledge have been explored (Mahajan et al., 2021; Gao et al., 2023; Feder et al., 2023).

**Semantic Corruptions.** In this paper, we explore the use of a different type of knowledge: corruptions of semantic features. Intuitively, imagine trying to predict the label from a corrupted input $T(\mathbf{x})$, where all semantic information has been removed. Any better-than-chance prediction provides us a window into the

---

*Corresponding email: aahlad@nyu.edu

nuisances, as it must rely on them. We will then use these obtained biased models to guide methods that we identify here as biased-model-based spurious-correlation avoiding methods (B-SCAMs).

**B-scams.** There is a class of methods in the literature that use predictions of a biased model to adjust for nuisances, and learn predictors that are free of spurious correlations. Among others, these include Just Train Twice (JTT) (Liu et al., 2021), EILL (Creager et al., 2021), Nuisance-Randomized Distillation (NURD) (Puli et al., 2022), and debiased focus loss (DFL), product of experts (POE) (Mahabadi et al., 2019). The key question arising from these works is *how can we obtain biased models?* In empirical studies, prior works on B-SCAMs either use annotations of the nuisance or an ERM-trained model over the training data as a placeholder for the biased model. The latter approach, based on an ERM-trained model, is successful if that model completely ignores semantic information. In practice, these heuristics are rather fragile. Annotations for nuisances are seldom available, and we lack a principled method to ascertain whether a model trained with ERM relies only on semantic features. Therefore, employing semantic corruptions could serve as a valuable alternative to these heuristics. We claim that semantic corruptions offer a principled and useful approach to obtaining biased models.

Semantic corruptions $T(\mathbf{x})$ must strike a delicate balance between removing semantic information and preserving nuisances. For example, if $T(\mathbf{x})$ replaces all pixels in an image with random noise, it corrupts semantics while simultaneously erasing all information about the nuisances. An ideal $T(\mathbf{x})$ would isolate nuisances by targeting only the semantic information in the input, e.g., by in-painting the animal for the task of classifying cows and penguins. Implementing such ideal corruptions is unrealistic, as they are task-specific and may require human annotations of the semantic features; e.g., one can segment the objects in every image. Doing so for all classification problems is extremely laborious. In tasks like NLI, it is unclear even *how* to annotate semantics, as they do not correspond to simple features like subsets of words. In summary, after outlining the desired characteristics of semantic corruptions, we define corruptions that are beneficial across multiple tasks and do not require human annotation. Our contributions are as follows:

1. Show that acquiring additional knowledge beyond a labeled dataset is necessary for effectively learning robust models (theorem 1). Then, in proposition 1, we formalize sufficient conditions under which additional knowledge in the form of a semantic corruption enables B-SCAMs to learn robust models.

2. Develop multiple semantic corruptions for object recognition and natural language inference. These include patch randomization, n-gram randomization, frequency filtering, and intensity filtering. Then, we situate existing procedures, such as region-of-interest masking and premise masking, under the umbrella of semantic corruptions.

3. Empirically, we demonstrate that any semantic corruption can power any B-SCAM. The corruption-powered versions of these methods outperform ERM on out-of-distribution (OOD) generalization tasks like Waterbirds, cardiomegaly detection from chest X-rays, and NLI. Corruption-powered NURD, DFL, and POE achieve performance similar to said methods run with extra observed nuisance variables. Corruption-powered JTT outperforms vanilla JTT.

## 2 Biased-model-based spurious-correlation avoiding methods

A spurious correlation is a relationship between the covariates $\mathbf{x}$ and the label $\mathbf{y}$ that changes across settings like time and location (Geirhos et al., 2020). The features whose relationship with the label changes are called nuisances. With a vector of nuisances $\mathbf{z}$, let $p_{tr}(\mathbf{y}, \mathbf{z}, \mathbf{x}), p_{te}(\mathbf{y}, \mathbf{z}, \mathbf{x})$ be the training and test distributions.

**Achieving robustness to spurious correlations requires additional knowledge.** In the presence of spurious correlations, the training distribution $p_{tr}$ may not equal the test distribution $p_{te}$. Without further assumptions, no algorithm that only sees data from $p_{tr}(\mathbf{y}, \mathbf{x})$ can produce a predictor that works well on $p_{te}$. To achieve generalization when $p_{te} \neq p_{tr}$, work in the OOD generalization literature assumes a relationship between the training and test distributions. We follow the work of Makar et al. (2022); Puli et al. (2022) and assume that only nuisance-label relationships — i.e. the conditional $\mathbf{z} \mid \mathbf{y}$ — changes between training and test. Formally, we let $p_{tr}, p_{te}$ come from a family of distributions whose members have different nuisance-label relationships but share the same relationship between the label and the semantics $\mathbf{x}^*$:

**Definition 1.** *(Nuisance-varying family with semantic features* $\mathbf{x}^*$ *([Makar et al., 2022](); [Puli et al., 2022]()))*

$$\mathcal{F} = \{p_D \;:\; p_D(\mathbf{y}, \mathbf{z}, \mathbf{x}^*, \mathbf{x}) = p(\mathbf{y}, \mathbf{x}^*)\, p_D(\mathbf{z} \mid \mathbf{y})\, p(\mathbf{x} \mid \mathbf{z}, \mathbf{x}^*)\}. \tag{1}$$

Many common tasks in OOD generalization, including some from [section 4](), fit this definition. For example, in classifying natural images, the background type is the nuisance $\mathbf{z}$ and its relationship to the label can change across places, each corresponding to a different member of $\mathcal{F}$. The animal shape however is made of semantic features $\mathbf{x}^*$ that are related to the label in the same way across places. Like in this example, we assume that the semantic features $\mathbf{x}^*$ equal a function of the covariates $\mathbf{x}^* = e(\mathbf{x})$ almost surely under every $p_D \in \mathcal{F}$, but neither $\mathbf{x}^*$ nor $e(\cdot)$ are known. Finally, the semantics and nuisances together account for all the information that $\mathbf{x}$ has about $\mathbf{y}$, meaning $\mathbf{x} \perp\!\!\!\perp_{p_D} \mathbf{y} \mid \mathbf{x}^*, \mathbf{z}$.

Building models that are robust to a shifting nuisance-label relationship relies on additional knowledge, such as nuisance annotations, in the training data ([Sagawa et al., 2019](); [Veitch et al., 2021](); [Makar et al., 2022](); [Puli et al., 2022](); [Yao et al., 2022]()). Given knowledge of $\mathbf{z}$, work like ([Makar et al., 2022](); [Puli et al., 2022]()) estimate a distribution, denoted $p_\perp$, under which the label and nuisance are independent ($\mathbf{y} \perp\!\!\!\perp_{p_\perp} \mathbf{z}$): $p_\perp(\mathbf{y}, \mathbf{x}) = \int_{z, x^*} p(\mathbf{y}, \mathbf{x}^* = x^*) p_{tr}(\mathbf{z} = z) p(\mathbf{x} \mid \mathbf{z} = z, \mathbf{x}^* = x^*) dz dx^*$. Following [Puli et al. (2022)](), we call $p_\perp$ the *nuisance-randomized distribution*. The model $p_\perp(\mathbf{y} = 1 \mid \mathbf{x})$ achieves the lowest risk on any member of the family $\mathcal{F}$ amongst the set of risk-invariant models; see Proposition 1 ([Makar et al., 2022]()). However, even when $p_{tr}, p_{te} \in \mathcal{F}$ and optimal risk-invariant predictors can be built with nuisances, *it is impossible to always beat random chance when given data* $\{\mathbf{y}, \mathbf{x}\} \sim p_{tr}$:

**Theorem 1.** *For any learning algorithm, there exists a nuisance-varying family $\mathcal{F}$ where predicting with $p_\perp(\mathbf{y} = 1 \mid \mathbf{x})$ achieves* 90% *accuracy on all members such that given training data $\mathbf{y}, \mathbf{x}$ from one member $p_{tr} \in \mathcal{F}$, the algorithm cannot achieve better accuracy than* 50% *(random chance) on some $p_{te} \in \mathcal{F}$.*

The proof is in [appendix A]() and proceeds in two steps. With $\mathrm{ACC}_h(p)$ as the expected accuracy of a model $h$ on distribution $p$, the first step of the proof defines two nuisance-varying families $\mathcal{F}_1, \mathcal{F}_2$ such that no single model can perform well on both families simultaneously; any $h(\mathbf{x})$ for which $\mathrm{ACC}_{p_1}(h) > 50\%$ for all $p_1 \in \mathcal{F}$ will have that $\mathrm{ACC}_{p_2}(h) < 50\%$ for some $p_2 \in \mathcal{F}_2$ and vice-versa. The second step shows that the two families $\mathcal{F}_1, \mathcal{F}_2$ have a member that has the same distribution over $\mathbf{y}, \mathbf{x}$; letting the training data come from this distribution means that any learning algorithm that returns a performant model — one that beats 50% accuracy – on one family must fail to return a performant model on the other. Next, we discuss different methods that use additional knowledge beyond $\mathbf{y}, \mathbf{x}$ to build robust predictors.

## 2.1 Biased-model-based spurious-correlation avoiding methods.

We focus on methods that correct models using knowledge of nuisances or where they might appear in the covariates ([Mahabadi et al., 2019](); [Liu et al., 2021](); [Puli et al., 2022]()). We first establish that the common central part in these methods is a model that predicts the label using nuisances, which we call the *biased model*; due to this commonality, we call these biased-model-based spurious-correlation avoiding methods (B-SCAMs). At a high level, a B-SCAM has two components. The first is a biased model that is built to predict the label by exploiting the nuisance-label relationship via extra knowledge or assumptions. The biased model is then used to guide a second model to predict the label without relying on nuisances.

We briefly summarize the different B-SCAMs here, differentiated by the additional knowledge they use to build biased models. The differences between the methods are summarized in [table 1](). We give details for NURD here and defer algorithmic details about the rest to [appendix B]().

**Biased models from knowledge of the nuisances.** The first category of B-SCAMs from [Mahabadi et al. (2019)](); [Puli et al. (2022)]() *assumes additional knowledge in the form of nuisance annotations* $\mathbf{z}$. For example, in NLI — where the goal is determining if a premise sentence entails a hypothesis — ([Mahabadi et al., 2019]()) compute the fraction of words shared between the hypothesis and the premise for each sample in the training data and use this as one of the nuisance features in building the biased model. The biased model in NURD, POE, DFL is learned by predicting the label from the nuisance annotations in the training data to estimate $p_{tr}(\mathbf{y} \mid \mathbf{z})$. Using nuisance annotations, [Puli et al. (2022)](); [Makar et al. (2022)]() use the model $p_{tr}(\mathbf{y} \mid \mathbf{z})$ as

**Table 1:** Summary of NURD, JTT, POE, and DFL. Each method approximates the biased model: $p_{tr}(\mathbf{y} \mid \mathbf{z})$. This table describes the different biased models, their names, how they are built.

| Method | Name | What the biased model is | Assumptions/Knowledge |
|---|---|---|---|
| JTT | Identification model | $p_{tr}(\mathbf{y} \mid \mathbf{x})$ learned via ERM | ERM learns biased models. |
| POE/DFL | Biased model | $p_{tr}(\mathbf{y} \mid \mathbf{z})$ learned via ERM | $\mathbf{z}$ from domain-knowledge. |
| NURD | Weight model | $p_{tr}(\mathbf{y} \mid \mathbf{z})$ learned via ERM | $\mathbf{z}$ from domain-knowledge. |

the biased model to define importance weights and minimize risk w.r.t a distribution $p_{\perp}$ obtained as

$$p_{\perp}(\mathbf{y}, \mathbf{z}, \mathbf{x}) = p_{tr}(\mathbf{y})p_{tr}(\mathbf{z})p(\mathbf{x} \mid \mathbf{y}, \mathbf{z}) = \frac{p(\mathbf{y})}{p_{tr}(\mathbf{y} \mid \mathbf{z})}p_{tr}(\mathbf{z})p_{tr}(\mathbf{y} \mid \mathbf{z})p(\mathbf{x} \mid \mathbf{y}, \mathbf{z}) = \frac{p(\mathbf{y})}{p_{tr}(\mathbf{y} \mid \mathbf{z})}p_{tr}(\mathbf{y}, \mathbf{z}, \mathbf{x}).$$

The second step in NURD (Puli et al., 2022) trains a model to predict $\mathbf{y}$ from a representation $r(\mathbf{x})$ on data from $p_{\perp}$ such that $\mathbf{z} \perp\!\!\!\perp_{p_{\perp}} \mathbf{y} \mid r(\mathbf{x})$; this step is called distillation. Due to $\mathbf{y} \perp\!\!\!\perp_{p_{\perp}} \mathbf{z}$, learning in $p_{\perp}$ avoids features that depend only on the nuisance and due to $\mathbf{z} \perp\!\!\!\perp_{p_{\perp}} \mathbf{y} \mid r(\mathbf{x})$, distillation avoids features that are mixed functions of the label and the nuisance (e.g. $\mathbf{x}_1 = \mathbf{y} + \mathbf{z}$). With these insights, NURD builds models of the form $p_{\perp}(\mathbf{y} \mid r(\mathbf{x}))$ that are most informative of the label. Mechanically, NURD's distillation solves this:

$$\max_{\theta, \gamma} \mathbf{E}_{p_{\perp}} \log p_{\theta}(\mathbf{y} \mid r_{\gamma}(\mathbf{x})) - \lambda \mathbf{I}_{p_{\perp}}(\mathbf{y}; \mathbf{z} \mid r_{\gamma}(\mathbf{x})).$$

Puli et al. (2022) show that such models are best in a class of predictors with lower bounds on performance. The mutual information above is zero when $\mathbf{y} \perp\!\!\!\perp_{p_{\perp}} \mathbf{z} \mid \mathbf{x}$; this condition holds for semantic corruptions as we discuss in appendix B. Thus, we run the distillation step as importance-weighted ERM on the training data.

Mahabadi et al. (2019) consider two methods to train a biased model and a base predictive model jointly to make the base model predict without relying on the biases. They propose 1) POE, where the loss is the sum of the `log` loss of the two models and 2) DFL, where the biased model is used to weight the cross-entropy loss for the base model. For both methods, Mahabadi et al. (2019) build a biased model as $p_{tr}(\mathbf{y} \mid \mathbf{z})$. Intuitively, the base model focuses on classifying samples that the biased model misclassifies. The methods fine-tune a BERT model (Devlin et al., 2019) and do not propagate the gradients of the biased model to update the common parameters (token embeddings).

**Biased models from assumptions on erm-trained models.** The second category of B-SCAMs like LFF (Nam et al., 2020), UMIX (Han et al., 2022), and JTT (Liu et al., 2021) require *additional knowledge that vanilla ERM builds a biased model that exploits the nuisance-label relationship.* Given such a model, these works use it to reduce a second model's dependence on the nuisance. We focus on JTT (Liu et al., 2021) which aims to build models robust to group shift, where the relative mass of a fixed set of disjoint groups of the data changes between training and test times. The groups here are subsets of the data defined by a pair of values of discrete label and nuisance values. While JTT works without relying on training group annotations, i.e. without nuisances, it assumes ERM's missclassifications are because of a reliance on the nuisance. JTT first builds an "identification" model via ERM to isolate samples that are misclassified. Then, JTT trains a model via ERM on data with the loss for the misclassified samples upweighted (by constant $\lambda$). The epochs to train the identification model and the upweighting constant are hyperparameters that require tuning using group annotations (Liu et al., 2021).

**The commonality of a biased model.** The central part in NURD, DFL, POE, and JTT is a model that predicts the label using nuisances, like $p_{tr}(\mathbf{y} \mid \mathbf{z})$, which we call the biased model as in He et al. (2019). The predictive models in each B-SCAM are guided to not depend on nuisances used by the biased model. While B-SCAMs reduce dependence on nuisances, they build biased models using additional nuisance annotations or require assumptions that ERM-trained models predict using the nuisance. In the next section, we describe an alternative: corrupt semantic information with data augmentations to construct biased models.

## 3 Out-of-distribution generalization via Semantic Corruptions

The previous section summarized how biased models can be built in B-SCAMs using either direct knowledge of nuisances or knowledge that ERM-trained models rely on the nuisances. We now introduce semantic

corruptions and show how they enable building biased models. Semantic corruptions are transformations of the covariates that do not retain any knowledge of the semantics, except what may be in the nuisance $\mathbf{z}$:

**Definition 2** (Semantic Corruption). *A semantic corruption is a transformation of the covariates $T(\mathbf{x}, \boldsymbol{\delta})$, where $\boldsymbol{\delta}$ is a random variable such that $\boldsymbol{\delta} \perp\!\!\!\perp (\mathbf{y}, \mathbf{z}, \mathbf{x}, \mathbf{x}^*)$, if*

$$\forall p_D \in \mathcal{F} \quad T(\mathbf{x}, \boldsymbol{\delta}) \perp\!\!\!\perp_{p_D} \mathbf{x}^* \mid \mathbf{z}.$$

Here, we characterize conditions under which biased models built from semantic corruptions could be used to estimate robust models. As discussed in section 2, $p_\perp(\mathbf{y} \mid \mathbf{x})$ is the optimal risk-invariant predictor, and is the target of ERM when predicting the label $\mathbf{y}$ from $\mathbf{x}$ under the nuisance-randomized distribution $p_\perp$. NURD estimates this distribution as part of the algorithm, and methods like JTT aim to approximate $p_\perp$, for example, upweighting samples mis-classified by a model that relies on $\mathbf{z}$ to predict $\mathbf{y}$. We compare $p_\perp$ which is obtained by breaking the nuisance-label relationship against the distribution obtained by breaking the relationship between the label and the data augmentation :

$$p_\perp(\mathbf{y}, \mathbf{x}) = \int_z \frac{p_{tr}(\mathbf{y})}{p_{tr}(\mathbf{y} \mid \mathbf{z} = z)} p_{tr}(\mathbf{y}, \mathbf{z} = z, \mathbf{x}), \qquad p_T(\mathbf{y}, \mathbf{x}) = \int_\delta p(\boldsymbol{\delta} = \delta) \frac{p_{tr}(\mathbf{y})}{p_{tr}(\mathbf{y} \mid T(\mathbf{x}, \delta))} p_{tr}(\mathbf{y}, \mathbf{x}) d\delta.$$

We show here that the $L_1$ distance between $p_\perp(\mathbf{y}, \mathbf{x})$ and $p_T(\mathbf{y}, \mathbf{x})$ is controlled by an $L_2$-distance between the biased models built from the nuisance and the data augmentations respectively:

**Proposition 1.** *Let $T : \mathbf{X} \times \mathbf{R}^d \rightarrow \mathbf{X}$ be a function. Assume the r.v. $p_{tr}(\mathbf{y} \mid T(\mathbf{x}, \boldsymbol{\delta}))^{-1}$ has a bounded second moment under the distribution $p_\perp(\mathbf{y}, \mathbf{z}, \mathbf{x})p(\boldsymbol{\delta})$, and that $p_{tr}(\mathbf{y} \mid T(\mathbf{x}, \boldsymbol{\delta}))$ and $p_{tr}(\mathbf{y} \mid \mathbf{z})$ satisfy*

$$\mathbb{E}_{p_\perp(\mathbf{y}, \mathbf{z}, \mathbf{x})p(\boldsymbol{\delta})} p_{tr}(\mathbf{y} \mid T(\mathbf{x}, \boldsymbol{\delta}))^{-2} \leq m^2, \qquad \mathbb{E}_{p_\perp(\mathbf{y}, \mathbf{z}, \mathbf{x})p(\boldsymbol{\delta})} |p_{tr}(\mathbf{y} \mid T(\mathbf{x}, \boldsymbol{\delta})) - p_{tr}(\mathbf{y} \mid \mathbf{z})|^2 = \epsilon^2.$$

*Then, the $L_1$ distance between $p_\perp(\mathbf{y}, \mathbf{x})$ and $p_T(\mathbf{y}, \mathbf{x})$ is bounded: $\|p_\perp(\mathbf{y}, \mathbf{x}) - p_T(\mathbf{y}, \mathbf{x})\|_1 \leq m\epsilon$. For a semantic corruption that also satisfies $\mathbf{y} \perp\!\!\!\perp_{p_{tr}} \mathbf{z} \mid T(\mathbf{x}, \boldsymbol{\delta})$ the inequalities hold with $\epsilon = 0$.*

If $\epsilon = 0$, $p_T(\mathbf{y}, \mathbf{x}) = p_\perp(\mathbf{y}, \mathbf{x})$ which means that almost surely the conditionals match $p_\perp(\mathbf{y} \mid \mathbf{x}) = p_T(\mathbf{y} \mid \mathbf{x})$. Then, as $p_\perp(\mathbf{y} \mid \mathbf{x})$ is the optimal risk-invariant predictor, so is $p_T(\mathbf{y} \mid \mathbf{x})$. More generally, standard domain adaptation risk bounds that are controlled by the total variation distance between source and target (Ben-David et al., 2010, Theorem 1) bound the risk of a model under $p_\perp$ using the $L_1$ bound $m\epsilon$ — which upper bounds the total variation — and the risk under $p_T$.

Without nuisance annotations, one cannot test whether estimate the $L_2$-distance between the two biased models $p_{tr}(\mathbf{y} \mid \mathbf{z})$ and $p_{tr}(\mathbf{y} \mid T(\mathbf{x}, \boldsymbol{\delta}))$ in proposition 1. This distance can be large when a transformation $T(\mathbf{x}, \boldsymbol{\delta})$ retains semantic information. To avoid, we turn to a complementary source of knowledge: semantic features. Using this knowledge, we design families of data augmentations that corrupt the semantic information in $\mathbf{x}$ to construct semantic corruptions. Focusing on two popular OOD tasks, object recognition and NLI, we use **only semantic knowledge** to build corruptions that retain some aspects of the covariates. Biased models built on such corruptions will depend on any retained nuisances; more retained nuisances mean better biased models.

### 3.1 Semantic corruptions via permutations

We first build corruptions when semantics appear as global structure. We give an intuitive example for such global semantics. Consider the waterbirds dataset from Sagawa et al. (2019) with waterbirds and landbirds appearing predominantly on backgrounds with water and land respectively. Semantic features like the wing shape and the presence of webbed feet are corrupted by randomly permuting small patches. See fig. 1a. Formally, given subsets of the covariates $\mathbf{x}_1, \cdots \mathbf{x}_k$ extracted in an order, global semantics $e(\mathbf{x}_1, \cdots, \mathbf{x}_k)$ change with the order of extraction. Formally, with a random permutation $\pi \sim q(\pi)$ and recalling that semantics are $\mathbf{x}^* = e(\mathbf{x})$, the information about semantics is lost after permutation: $\forall p_D, \mathbf{I}_{p_D, q(\pi)}(\mathbf{x}^*; e(\mathbf{x}_{\pi(1)}, \cdots \mathbf{x}_{\pi(k)}))) = 0$.

We give an example of a semantic corruption with global semantics. Consider distributions $\{p_D\}_{D \in \mathbf{R}}$ with different nuisance-label relationships. With $\mathcal{U}$ as the uniform distribution over $\{1, 2, 3\}$, and $\mathcal{N}$ as the normal distribution, $p_D(\mathbf{y}, \mathbf{z}, \mathbf{x})$ corresponds to $\mathbf{y} \sim \mathcal{U}$, $\mathbf{z} \sim \mathcal{N}(D\mathbf{y}, 1)$, and $\mathbf{y}$ selecting a configuration of $\mathbf{x}$

$$\mathbf{y} = 1 \implies \mathbf{x} = [-\mathbf{z}, \mathbf{z}, \mathbf{z}], \qquad \mathbf{y} = 2 \implies \mathbf{x} = [\mathbf{z}, -\mathbf{z}, \mathbf{z}], \qquad \mathbf{y} = 3 \implies \mathbf{x} = [\mathbf{z}, \mathbf{z}, -\mathbf{z}]$$

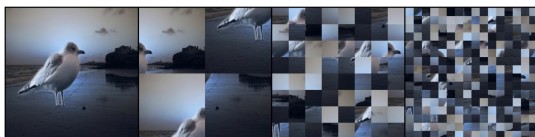

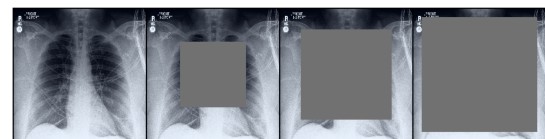

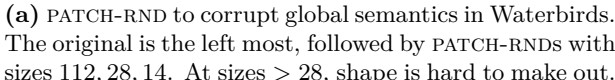

**(a)** PATCH-RND to corrupt global semantics in Waterbirds. The original is the left most, followed by PATCH-RNDs with sizes $112, 28, 14$. At sizes $> 28$, shape is hard to make out.

**(b)** Masking to corrupt semantics in chest X-rays. The original is the left most, followed by ROI-MASK of size $112, 154, 196$. At sizes $> 154$, the heart is blocked out.

**Figure 1:** Semantic corruptions of Waterbirds via PATCH-RND and chest X-rays via ROI-MASK.

The index of the negated coordinate is the semantic feature $\mathbf{x}^*$ that equals $\mathbf{y}$ and computing it requires comparing coordinates: $\mathbf{y} = 1$ if $\mathbf{x}_2\mathbf{x}_3 > 0$, $\mathbf{y} = 2$ if $\mathbf{x}_1\mathbf{x}_3 > 0$, and $\mathbf{y} = 3$ otherwise. In words, the semantic feature is global. However, $\mathbf{z} = \mathbf{x}_1 + \mathbf{x}_2 + \mathbf{x}_3$ is determined regardless of where the negative sign is, i.e. it is not global. A random permutation $T(\mathbf{x}, \boldsymbol{\delta})$ of the coordinates in $\mathbf{x}$ is thus a semantic corruption: as $T(\mathbf{x}, \boldsymbol{\delta})$ permutes the location of the negation, $T(\mathbf{x}, \boldsymbol{\delta}) \mid \mathbf{y}, \mathbf{z}$ is distributed identically to $T(\mathbf{x}, \boldsymbol{\delta}) \mid \mathbf{z}$. In turn, $T(\mathbf{x}, \boldsymbol{\delta}) \perp\!\!\!\perp \mathbf{y} \mid \mathbf{z}$. Further, the product of the three coordinates of $T(\mathbf{x}, \boldsymbol{\delta})$ determines $\mathbf{z}$: $(\Pi_{i \in \{1,2,3\}} T(\mathbf{x}, \boldsymbol{\delta})_i)^{1/3} = -\mathbf{z}$. Thus, $T(\mathbf{x}, \boldsymbol{\delta})$ determines $\mathbf{z}$ and $\mathbf{y} \perp\!\!\!\perp \mathbf{z} \mid T(\mathbf{x}, \boldsymbol{\delta})$. These two independencies imply that $\epsilon = 0$ in proposition 1. Then, biased models from $T(\mathbf{x})$ are as good as ones from $\mathbf{z}$. Next, we give corruptions for global semantics in vision and language tasks, that retain non-global features.

**Patch randomization.** Object recognition tasks where the object is a shape that can satisfy the global property. For illustration, consider differentiating cows and penguins in natural images; here, shape is a global semantic feature that structures multiple patches. Permuting patches via *patch randomization (PATCH-RND)*, like in fig. 1a, corrupts global semantics.

**N-gram randomization.** Tasks like natural language inference (NLI) — where the goal is determining if a premise sentence entails a hypothesis — satisfy the global-semantics property. Consider this example: the sentence "Bob speaks but Jon does not" contradicts "Jon speaks but Bob does not" but entails "Bob speaks". The meaning is inferred from a global structure on the words and the order they appear in. Here, randomizing the order of the words corrupts the semantics: For example, one randomized order of the sentence "Jon speaks but Bob does not" is "Bob speaks but Jon does not"; the former entails "Jon speaks" but the latter contradicts it. We randomize the order by permuting different $n$-grams in each sentence; we call this *n-gram randomization (NGRAM-RND)*.

### 3.2 Semantic corruptions via masking

The second corruption we build focuses on cases where certain subsets of the covariates are necessary part of semantics. Masking, by removing that subset or setting it to a constant, corrupts semantics. Formally, we corrupt the semantics by replacing subsets $\mathbf{x}_S$ with a value that is out of support: for example, in images where pixels lie in $(0, 1)$, we corrupt $\mathbf{x} = [\mathbf{x}_S, \mathbf{x}_{-S}]$ as $\mathbf{x}_{\text{corrupted}} = [0 * \mathbf{x}_S, \mathbf{x}_{-S}]$. As an illustrative example, consider a family $\mathcal{F} = \{p_D\}_{D \in R}$ with varying nuisance-label relationships. With $\mathbf{a}, \mathbf{b}$ being uniform binary random variables, $\mathbf{e}(\rho)$ as the exponential distribution with parameter $\rho$, and $s_+(u) = \log(1 + \exp(u))$ as softplus, sample from $p_D(\mathbf{y}, \mathbf{z}, \mathbf{x})$ as: $\quad \mathbf{y} = \mathbf{a} \oplus \mathbf{b}, \quad \mathbf{z} \sim \mathbf{e}(s_+(D * (2\mathbf{y} - 1))), \quad \mathbf{x} = [(2\mathbf{a} - 1)\mathbf{z}, (2\mathbf{b} - 1)\mathbf{z}]$.

For such a family, we show that masking out coordinate $\mathbf{x}_1$ is a semantic corruption: $T(\mathbf{x}) = [0, \mathbf{x}_2]$ satisfies $T(\mathbf{x}) \perp\!\!\!\perp \mathbf{y} \mid \mathbf{z}$ and $T(\mathbf{x}) \not\!\perp\!\!\!\perp \mathbf{z}$. First, due to $\mathbf{y}$ being computed as an XOR function of $\mathbf{a}, \mathbf{b}$, it holds that $\mathbf{b} \perp\!\!\!\perp \mathbf{y}$. Then, due to $\mathbf{z}$ only relying on $\mathbf{y}$ and exogenous noise, $\mathbf{b} \perp\!\!\!\perp (\mathbf{y}, \mathbf{z})$ which implies $\mathbf{b} \perp\!\!\!\perp \mathbf{y} \mid \mathbf{z}$. Given $\mathbf{z}$, $\mathbf{b}$ determines $\mathbf{x}_2$, so $\mathbf{b} \perp\!\!\!\perp \mathbf{y} \mid \mathbf{z} \implies \mathbf{x}_2 \perp\!\!\!\perp \mathbf{y} \mid \mathbf{z} \implies T(\mathbf{x}) \perp\!\!\!\perp \mathbf{y} \mid \mathbf{z}$. Further, $\|T(\mathbf{x})_2\| = \mathbf{z}$ which means $\mathbf{y} \perp\!\!\!\perp \mathbf{z} \mid T(\mathbf{x})$. These two independencies imply that $\epsilon = 0$ in proposition 1. Then, using $T(\mathbf{x})$ to build biased models is equivalent to building them with $\mathbf{z}$.

**ROI-masking for object recognition.** Semantics in images can often be localized to a region-of-interest (ROI). For example, in detecting cardiomegaly, the ROI is the chest where the heart resides. Masking out the ROI removes centrally located semantic information from the chest X-ray (fig. 1b). We call this *region-of-interest masking (ROI-MASK)*.

**Premise-masking for NLI.** Semantic features in NLI rely on the meanings of the premise and the hypothesis sentences: for example, the premise states the occurrence of an event ("Alice sat while Bob stood.")

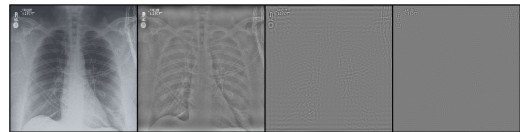

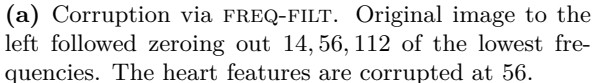

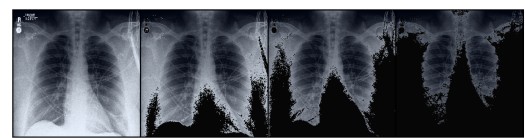

**(a)** Corruption via FREQ-FILT. Original image to the left followed zeroing out $14, 56, 112$ of the lowest frequencies. The heart features are corrupted at 56.

**(b)** Corruption via INT-FILT. Original image to the left followed by zeroing out pixels with intensities above the $80\%, 60\%, 40\%$. Heart features look corrupted at 40%.

**Figure 2:** Semantic corruptions of chest X-rays via FREQ-FILT and INT-FILT respectively.

which can entail ("Alice sat.") or contradict ("Bob sat.") the hypothesis. The information about the setup in the premise is therefore crucial to detect entailment or contradiction. If the context given by the premise is blocked out, the hypothesis sentence can predict the label only due to nuisances. Thus, masking the premise is a semantic corruption for NLI that retains hypothesis features; we call this *premise masking (PREM-MASK)*.

### 3.3 Semantic corruptions via frequency and intensity filters

PATCH-RND relies on differences in relative size and ROI-MASK relies on differences in spatial position. We consider two aspects of the image that are not spatial, frequency and pixel-intensity, and give corruptions for features that depend on these aspects. Semantics can appear as signals in a particular region of the frequency spectrum, or appear at a particular luminosity in the image. For example, consider detecting cardiomegaly from chest X-rays, where the heart appears as an object formed of bright pixels with little variation in intensity across the pixels; the latter suggests that the heart features are low-frequency signals.

This observation motivates corruptions along the axes of frequency and pixel-intensity: *frequency filtering (FREQ-FILT)* and *intensity filtering (INT-FILT)*. FREQ-FILT zeroes-out frequencies in the discrete fourier domain, while INT-FILT zero-out pixels based on their intensities. See fig. 2 for how FREQ-FILT and INT-FILT corrupt the heart region. FREQ-FILT and INT-FILT require characterizing semantic features in frequency and intensity spaces; this is in contrast to ROI-MASK that is based on characterizing the position in pixel space that the semantics occur in.

### 3.4 Using semantic corruptions in practice

For each method in table 1, we use a semantic corruption $T(\mathbf{x})$ in building a model $p_{tr}(\mathbf{y} \mid T(\mathbf{x}))$. For reweighting-NURD, we replace $p_{tr}(\mathbf{y} \mid \mathbf{z})$ with $p_{tr}(\mathbf{y} \mid T(\mathbf{x}))$, for DFL and POE, we replace the model $p_{tr}(\mathbf{y} \mid \mathbf{z})$ with $p_{tr}(\mathbf{y} \mid T(\mathbf{x}))$, and for JTT, we use $p_{tr}(\mathbf{y} \mid T(\mathbf{x}))$ as the identification model.

**Choosing the corruption parameter.** To corrupt with PATCH-RND, NGRAM-RND, and ROI-MASK, FREQ-FILT, one must select a size parameter and to corrupt with INT-FILT, one must specify an intensity threshold. For NURD, JTT, POE and DFL, we select corruption parameters with the same validation schemes used to select other hyperparameters in each respective paper. In practice, including the B-SCAMs run without semantic corruptions in the B-SCAM's validation scheme ensures a lower bound on performance. For example, for JTT, this inclusion yields a lower bound that corresponds to vanilla JTT's performance. We also report results for all corruption parameters in appendix C.3, showing that all semantic corruptions except INT-FILT are not sensitive to the parameters, and lead to models that outperform ERM.

## 4 Experiments

We study semantic corruptions in powering NURD (Puli et al., 2022), JTT (Liu et al., 2021), and POE and DFL (Mahabadi et al., 2019). To be faithful to the original evaluations of each method, we run them on tasks from their respective papers: NURD on waterbirds, JTT on waterbirds and NLI where the nuisance is the presence of a negation word, and POE and DFL on NLI evaluated on a challenging test dataset, HANS (McCoy et al., 2019). We run NURD on chest X-rays but focus on detecting cardiomegaly rather than the original pneumonia (Puli et al., 2022) because pneumonia detection even with known-nuisances is not performant. See appendix C for details and appendix C.3 for additional experiments investigating semantic corruptions.

**Methods, metrics and model selection.** For images, we corrupt semantics with PATCH-RND, a central ROI-MASK, FREQ-FILT, and INT-FILT. To show the value of semantic corruptions relative to existing data

augmentations, we also consider two baseline transformations of images. The first is random cropping (RAND-CROP) like in self-supervised learning (Bardes et al., 2021; Chen et al., 2020) where patches of random sizes are sampled, covering $\geq 0.08$ fraction of the image. The second is adding gaussian noise (GAUSS-NOISE). For text, we corrupt semantics with NGRAM-RND and PREM-MASK. We report the average test accuracy for every method. To be able to compare to what JTT is trained for in Liu et al. (2021), we report worst-group test accuracy for JTT. For each method, we compare the performance of the original method to that of the methods run with semantic corruption (including the baselines). For the corruption-powered versions, group annotations and nuisances are *unavailable* in the training data. Known-nuisance versions of POE, DFL, and NURD use direct knowledge of one or more nuisances during training. In choosing parameters and early stopping, like Liu et al. (2021) do with vanilla JTT, corruption-powered JTT uses validation group annotations. For the other methods, we follow validation schemes from the respective papers: for NURD we follow Puli et al. (2022) and use a validation set weighted to have independent nuisance and label, and for POE/DFL, we follow Mahabadi et al. (2019) and use a set of 1000 samples from the HANS training dataset.

## 4.1 Object recognition tasks

To be faithful to the original evaluations of each method, we evaluate JTT on waterbirds, and NURD on both waterbirds and detecting cardiomegaly; both tasks have images of size $224 \times 224 \times 3$. Both Puli et al. (2022) and Liu et al. (2021) use the raw waterbirds data from Sagawa et al. (2019), where the task is detecting the type of bird (water or land) from images where the background is a nuisance. Unlike Liu et al. (2021), Puli et al. (2022) process the waterbirds to get a different setup from Sagawa et al. (2019). To stay true to the original evaluations of the methods, we recreate the setups as described in their respective papers. For both tasks, we use PATCH-RND (of patch sizes $7, 14, 28, 56$), ROI-MASK (of mask sizes $112, 140, 168, 196$), FREQ-FILT (of high-pass filter sizes $196, 168, 140, 112$), and INT-FILT (of thresholds $0.1, 0.2, 0.3, 0.4$) as semantic corruptions. For GAUSS-NOISE, we use variances $0.01, 0.25, 1, 4$.

**Nurd on waterbirds.** For NURD, we recreate the waterbirds experiment from Puli et al. (2022) where the full waterbirds data from Sagawa et al. (2019) was subsampled into training, validation, and test datasets each with label balance. However, unlike Sagawa et al. (2019), the validation data comes from the same distribution as the training data. The training and validation datasets have 90% waterbirds on backgrounds with water and 90% landbirds on backgrounds with land. The test data has a flipped relationship. Known-nuisance NURD uses an additional label denoting the background-type as the nuisance.

**Table 2:** Mean and standard error of test accuracy across 10 seeds of NURD with semantic corruptions on waterbirds. *Known*-**z** NURD uses a label for the type of background as the nuisance. Consider the gap between ERM and known-nuisance NURD. NURD with semantic corruptions PATCH-RND, ROI-MASK, FREQ-FILT, and INT-FILT close $99\%, 99\%, 82\%, 99\%$ of the gap respectively. NURD with semantic corruptions outperforms ERM and NURD with RAND-CROP, GAUSS-NOISE.

| Method | test acc. |
|---|---|
| *Known*-**z** NURD | $87.2 \pm 1.0\%$ |
| PATCH-RND | $86.9 \pm 1.2\%$ |
| ROI-MASK | $86.9 \pm 1.7\%$ |
| FREQ-FILT | $83.5 \pm 1.1\%$ |
| INT-FILT | $86.9 \pm 1.1\%$ |
| RAND-CROP | $73.7 \pm 2.0\%$ |
| GAUSS-NOISE | $82.0 \pm 2.6\%$ |
| ERM | $68.0 \pm 1.9\%$ |

Table 2 gives results. Selecting hyperparameters using NURD's validation approach gives sizes 14 for PATCH-RND (86.9%), 196 for ROI-MASK (86.9%), 168 for FREQ-FILT (83.5%), and threshold 0.2 for INT-FILT (86.9%). Consider the gap between ERM and known-nuisance NURD. NURD with PATCH-RND, ROI-MASK, FREQ-FILT, and INT-FILT close $99\%, 99\%, 82\%, 99\%$ of the gap respectively. NURD with these semantic corruptions outperforms ERM (68.0%) and NURD with RAND-CROP (73.7%) and GAUSS-NOISE (82.0%). Additionally, in table 11 in appendix C, we give the results for all corruption parameters, showing that NURD with semantic corruptions is *insensitive to hyperparameters of the corruption* and outperforms ERM. In appendix C.1, we discuss how the baseline GAUSS-NOISE could close 80% of the gap between ERM and known-**z** NURD.

**JTT on waterbirds.** For JTT, we repeat the waterbirds experiment from Liu et al. (2021) which uses the original data from Sagawa et al. (2019). The training data has 95% waterbirds on backgrounds with water and 95% landbirds on backgrounds with land. Both the validation and test datasets have bird label independent of the background. The groups here are subsets of the data that correspond to a pair of values

of bird-type and background-type. Like vanilla JTT, we use group annotations in the validation data to compute worst-group error and early stop training when using PATCH-RND and ROI-MASK. The results for vanilla JTT are from our run using the optimal hyperparameters from Liu et al. (2021).

Table 3 shows the results. Selecting the corruption hyperparameters on the validation worst-group accuracy gives size 14 for PATCH-RND (89%), size 196 for ROI-MASK (88.2%), size 112 for FREQ-FILT (87.2%), and threshold 0.4 for INT-FILT (87.0%). JTT with these semantic corruptions outperforms ERM (72.0%), vanilla JTT (86.5%), and JTT with the baseline corruptions RAND-CROP (75%) and GAUSS-NOISE (71%). Additionally, table 13 shows that JTT with PATCH-RND and ROI-MASK outperforms JTT with the baseline corruptions and ERM at every patch/border-size.

**Table 3:** Test worst-group (WG) accuracies of JTT on waterbirds. JTT with semantic corruptions outperforms ERM, vanilla JTT, and JTT with baseline corruptions (RAND-CROP, GAUSS-NOISE).

| Method | test WG acc. |
|---|---|
| *Vanilla* JTT | 86.5% |
| PATCH-RND | 89.0% |
| ROI-MASK | 88.2% |
| FREQ-FILT | 87.2% |
| INT-FILT | 87.0% |
| RAND-CROP | 75.0% |
| GAUSS-NOISE | 71.0% |
| ERM | 72.0% |

**Nurd on detecting cardiomegaly** In chest X-ray classification, differences between hospitals, like the scanners used to produce the X-rays, are known to correlate thoracic conditions with non-physiological aspects in the image; for example, only some scanners render the air in the lungs in white (Zech et al., 2018). We consider the shape-based object recognition task of cardiomegaly (an irregularly sized heart) detection and, following Puli et al. (2022), construct a dataset from two chest X-ray datasets: chexpert (Irvin et al., 2019) and MIMIC (Johnson et al., 2019). The training and validation datasets have 90% cardiomegaly images from MIMIC and 90% healthy images from chexpert, while the test data has a flipped relationship. Known-nuisance NURD uses hospital identity as the nuisance.

See table 4 for results. Selecting the corruption parameters using NURD's validation approach gives size 14 for PATCH-RND (77%), size 196 for ROI-MASK (78.7%), size 168 for FREQ-FILT (76.0%), and threshold 0.1 for the INT-FILT (71.0%). Consider the gap between ERM and known-nuisance NURD. NURD with PATCH-RND, ROI-MASK, FREQ-FILT, and INT-FILT close $72\%, 82\%, 65\%, 35\%$ of the gap respectively. NURD with all semantic corruptions, outperforms ERM (65.3%) and NURD with the baselines GAUSS-NOISE (69%) and RAND-CROP (59.9%). Additionally, we report results for all corruptions in table 11 in appendix C showing that NURD with PATCH-RND and ROI-MASK *are insensitive to hyperparameters* and outperform ERM.

**Table 4:** Mean and standard error of test accuracy over 10 seeds of NURD on chest X-rays. *Known-$\mathbf{z}$* NURD uses the hospital as the nuisance. Consider the gap between ERM and known-$\mathbf{z}$ NURD. NURD with PATCH-RND, ROI-MASK, FREQ-FILT, and INT-FILT close $72\%, 82\%, 65\%, 35\%$ of the gap respectively. Except with INT-FILT, NURD with semantic corruptions outperforms ERM and NURD with baseline corruptions (RAND-CROP, GAUSS-NOISE).

| Method | test acc. |
|---|---|
| *Known-$\mathbf{z}$* NURD | $81.7 \pm 0.3\%$ |
| PATCH-RND | $77.0 \pm 1.2\%$ |
| ROI-MASK | $78.7 \pm 0.3\%$ |
| FREQ-FILT | $76.0 \pm 0.6\%$ |
| INT-FILT | $71.0 \pm 1.0\%$ |
| RAND-CROP | $59.9 \pm 2.1\%$ |
| GAUSS-NOISE | $69.0 \pm 1.9\%$ |
| ERM | $65.3 \pm 1.1\%$ |

### 4.2 Natural language inference (nli)

For methods POE, DFL, and JTT, we use the MNLI dataset (Williams et al., 2018) to fine-tune a BERT model. The evaluations of these methods in their respective papers have different nuisances and, consequently, different test sets. In accordance, we describe the setup and the results separately. We use NGRAM-RND (sizes $1, 2, 3, 4$) to produce nuisances for both setups.

**PoE and DFL** For POE and DFL, we report test accuracies on the HANS dataset McCoy et al. (2019) as in Mahabadi et al. (2019). HANS was created to test the reliance of models on three known nuisances: 1) lexical overlap, 2) subsequence match, and 3) constituent matching subtrees in the parse trees. Known-nuisance POE and DFL use exact knowledge of these nuisances. Table 5 gives the mean test accuracies over 4 seeds. For both DFL and POE, selecting the size hyperparameter based on the average accuracy on a small subset of the HANS training data (1000 samples) gives $n = 3$. With this size, POE achieves 66.7%, improving over POE with known nuisances (66.3%). DFL with NGRAM-RND of size 3, achieves 68.4%, closing 84% of the gap between ERM and known-nuisance DFL (69.3%).

POE and DFL with PREM-MASK (PM) close 33% and 28% of the gap between ERM and when the methods have knowledge of **z**. We expect the methods with NGRAM-RND do better than with PREM-MASK because the latter corrupts nuisances like lexical overlap between premise and hypothesis that HANS focuses on. Additionally, we give results for all *n*-gram sizes in table 10 in appendix C, showing that POE and DFL beat ERM for all *n*-gram sizes. Further, in appendix C.3.1, we evaluate POE and DFL models on the ANLI (Nie et al., 2019) dataset and counterfactually-augmented data (Kaushik et al., 2019) in tables 15 and 16.

**JTT**   For JTT, we repeat the NLI experiment from Liu et al. (2021), where the presence of a negation word in the hypothesis sentence is the nuisance. The groups here are subsets of the data that correspond to a value of the label and whether or not there is a negation word in the hypothesis. Vanilla JTT uses group annotations in the validation data to tune the hyperparameters and early stop training. For each *n*-gram size, we run JTT with NGRAM-RND for two values of the number of epochs of training for the identification model: 2, 3. Following the hyperparameter selection procedure from Liu et al. (2021), for each *n*-gram size, we give the results for the run with the higher validation worst-group accuracy. *Vanilla* JTT is run with the optimization hyperparameters from (Liu et al., 2021).

Table 6 gives the results. Selecting the size hyperparameter for NGRAM-RND using validation worst-group accuracy, like Liu et al. (2021) do for JTT, gives $n = 1$ with test worst-group accuracy of 74.3%, better than vanilla JTT's 71.3%. Additionally, table 14 shows that JTT using NGRAM-RND at *every* size or PREM-MASK performs better than both vanilla JTT (71.3%) and ERM (67.9%).

## 5   Related work

Biased-model-based spurious-correlation avoiding methods (B-SCAMs) like (Veitch et al., 2021; Clark et al., 2019; Puli et al., 2022; He et al., 2019; Makar et al., 2022) assume the nuisance is available as additional knowledge during training. Semantic corruptions offer a complementary approach to hand-crafting nuisances or obtaining auxiliary labels, by capturing nuisances that remain after corruption (e.g. non-global nuisances remain after PATCH-RND). B-SCAMs like LFF (Nam et al., 2020), UMIX (Han et al., 2022), and JTT (Liu et al., 2021) all rely on one crucial assumption: that ERM-training builds a biased model that exploits the nuisance and use it to reduce a second model's dependence on the nuisance. Each method trains the second model with weighted cross-entropy with higher weights for samples misclassified by the biased model; the methods differ in how they build the biased model and how they compute the weighted loss. The biased models learn to predict the label from the covariates. Such a model can also rely on the semantic features and upweighting its misclassified samples produces data with a different label-semantic relationship from the one in the training data. Models trained on such data are suboptimal on test data which has the same semantic relationship as the training data. Using semantic corruptions in these B-SCAMs reduces the biased model's reliance on the semantics and makes the second model rely more on the semantics; thus, B-SCAMs that rely on assumptions on ERM-trained models being biased achieve better performance when using semantic corruptions. The experiments in section 4 confirm this empirically: JTT with semantic corruptions improves over vanilla JTT.

Two instances of semantic corruptions, PREM-MASK and ROI-MASK, appear in earlier work (Mahabadi et al., 2019; He et al., 2019; Puli et al., 2022) but were designed using knowledge of where nuisances appear in the covariates. (Puli et al., 2022) used the borders of X-ray images as features that are related only to the scanner type (the nuisance), and not human physiology, to avoid spurious correlations in the detection of cardiomegaly. For NLI, Mahabadi et al. (2019) use knowledge that the test set was constructed from samples

**Table 5:** Mean and standard deviation of accuracies (over 4 seeds) on the HANS dataset. The results for POE and DFL that use known nuisances are given under *known*. POE with NGRAM-RND (NR) performs better than known-nuisance POE. DFL with (NR) closes 84% of the gap between ERM and known-nuisance DFL. POE and DFL with PREM-MASK (PM) close 33% and 28% of the gap between ERM and the respective method with known **z**.

| Method | HANS test acc. |
|---|---|
| POE, *known*-**z** | $66.3 \pm 0.6\%$ |
| POE, NR | $66.7 \pm 1.5\%$ |
| POE, PM | $64.5 \pm 1.9\%$ |
| DFL, *known*-**z** | $69.3 \pm 0.2\%$ |
| DFL, NR | $68.4 \pm 1.5\%$ |
| DFL, PM | $65.2 \pm 0.7\%$ |
| ERM | $63.6 \pm 1.1\%$ |

**Table 6:** Worst-group and avg. test accuracies of JTT on NLI. JTT with PREM-MASK (PM) and NGRAM-RND (NR) outperforms vanilla JTT and ERM.

| | Worst-group | Avg. |
|---|---|---|
| *Vanilla* JTT | 71.3% | 79.1% |
| JTT + PM | 72.1% | 79.9% |
| JTT + NR | 74.3% | 79.7% |
| ERM | 67.9% | 82.4% |

misclassified by a model that relies on the hypothesis alone to build a biased model using the hypothesis sentence. These are special cases of ROI-MASK and PREM-MASK from section 3.2 repsectively. Our work widely generalizes the observations from the papers above by formally defining and further realizing the abstraction of semantic corruptions in several instances and across applications.

Bahng et al. (2020) uses CNNs with small receptive fields (RFs), to capture non-global nuisances. However, their method is typically limited to very small filters because at size 3x3, deep neural networks like VGG detect global semantics like shapes. In contrast, the size choice in PATCH-RND has no bearing on the choice of the model; we used default vision models. Bras et al. (2020) automatically identify and remove examples with nuisances using adversarial filtering, but risk removing genuinely easy examples. Qin et al. (2021) work solely with vision transformers and point out that nuisances are the only reason labels can be predicted from transformations akin to patch-randomized images. They propose to encourage transformers to have predictions and representations of the original images be dissimilar from those of patch-randomized ones. In contrast, our work applies to general flexible models and shows that semantic corruptions can be used to break the label's relationship with nuisances in the original images.

Yao et al. (2022); Gao et al. (2023) use additional knowledge about nuisances or environments to corrupt nuisances in the covariates, Yao et al. (2022) corrupt nuisances in the covariates via the Mixup (Zhang et al., 2017) of samples from different domains that share a label. Gao et al. (2023) directly randomize nuisances; for example, in detecting animals in their natural habitats, they place segmented animal foregrounds onto random habitat backgrounds. Unlike these methods, we design semantic corruptions using the complementary knowledge about semantics, which can be available even without knowledge about nuisances. Clark et al. (2019); Li and Vasconcelos (2019) construct nuisances in the training stage using prior knowledge: for example, (Clark et al., 2019) uses the first token of the hypothesis as a nuisance for a synthetic NLI task which was created to have the first token be spuriously correlated with the label. Another example is the VQA task where the question-type is used as the nuisance. The constructed nuisances are then used to build biased (or bias-only) models, or construct per-sample weights to de-bias the loss. In contrast, we use knowledge about semantics to corrupt them; for example, the order of the words is a semantic feature that is corrupted by randomizing the order. This construction does not use knowledge of the nuisance.

Sinha et al. (2021) use techniques like PATCH-RND to restrict supports in self-supervised learning and generative modeling. Carlucci et al. (2019) use PATCH-RND images to encourage a model to recover semantic structure. In contrast, we use PATCH-RND to corrupt semantics and build biased models that rely on the nuisances, which help build predictive models that avoid reliance on nuisances. We focus on corrupting semantic features using simple procedures (like permuting, masking, filtering) while papers (Kaushik et al., 2019; Teney et al., 2020; Feder et al., 2022; Kaushik et al., 2020; Eisenstein, 2022; Wang and Culotta, 2021; 2020) focus on perturbing semantic features while keeping other features the same. These transformations produce examples of different labels, and are called counterfactuals. These examples are used to counterfactually augment the training data (Kaushik et al., 2019). Constructing counterfactuals can be hard. Works like (Kaushik et al., 2019; Teney et al., 2020; Feder et al., 2022; Kaushik et al., 2020) rely on humans to create counterfactuals because it is difficult to automate semantic perturbation without changing nuisances. For example, consider classifying dogs versus cats. Creating a dog that looks like a specific cat is much harder than removing the cat from the image by masking out those pixels.

Methods like (Wang and Culotta, 2021; 2020) construct counterfactuals automatically, but require additional knowledge of how nuisances appear in the text. For example, Wang and Culotta (2021) matches sentences that have opposite labels while sharing most words. The non-shared words would then be considered semantic. Techniques like the matching one above from Wang and Culotta (2020) are unrealistic beyond the task of sentiment classification. For example, consider the label of entailment or contradiction in NLI. Data samples with entailment as the label that contain negation words are rare. This makes it hard to find a good counterfactual for data samples labeled with contradiction. Further, matching is difficult in other modalities, like images, where covariates are continuous or high-dimensional and live in spaces where metrics are unclear.

## 6 Discussion

We study the use of semantic knowledge in models robust to spurious correlations. In theorem 1, we show that additional knowledge is necessary to achieve OOD generalization even when the training and test

distributions are coupled in a nuisance-varying family. Then, proposition 1 shows that a biased model built from a transformation of the covariates $T(\mathbf{x}, \boldsymbol{\delta})$ — that is $p_{tr}(\mathbf{y} \mid T(\mathbf{x}, \boldsymbol{\delta}))$ — can power B-SCAMs to avoid nuisances if the biased model $p_{tr}(\mathbf{y} \mid T(\mathbf{x}, \boldsymbol{\delta}))$ is close to $p_{tr}(\mathbf{y} \mid \mathbf{z})$ in $L_2$ distance. There are two scenarios where this distance is large: the transformation does not corrupt semantics and it corrupts nuisances. We use knowledge of the semantics to design semantic corruptions to avoid the first scenario. *Since we work without nuisances*, to avoid the second scenario — that is to choose $T(\mathbf{x}, \boldsymbol{\delta})$ that retain nuisances — we use standard validation schemes in B-SCAMs. Using semantic corruptions, practitioners can run different kinds of B-SCAMs (NURD, JTT, DFL, POE). Corruption-powered methods like NURD and DFL perform close to how they would with known nuisances. For methods like JTT, the corruption-powered versions perform better than their vanilla versions that rely on ERM on the raw covariates to yield nuisances.

**Limitations.** The quality of any semantic corruption, and thus the quality of the results, depends on the extent to which semantics are destroyed and nuisances are retained. PATCH-RND and NGRAM-RND are built to corrupt global semantics, and therefore are most suitable for when the nuisances are local. ROI-MASK corrupts semantics in the ROI and PREM-MASK corrupts the semantic context in the premise; these are most suitable for when nuisances lie outside the region-of-interest (ROI) or in the hypothesis respectively. Finally, FREQ-FILT and INT-FILT corrupt semantics in particular parts of the frequency and intensity spectrum, and are most suitable for when the nuisances and semantics lie in separate parts of the spectra. Knowledge about the kind of nuisances present in a dataset can lead to better choices of semantic corruptions. Alternatively, one could use standard validation schemes to select a corruption, like we do in section 4.

When applied blindly, the procedures we describe may retain semantics or corrupt nuisances. PATCH-RND and NGRAM-RND may corrupt global nuisances and retain local semantics, ROI-MASK and PREM-MASK may corrupt nuisances that occur in the same region as the semantics, and FREQ-FILT and INT-FILT may corrupt both semantics and nuisances if they appear at similar frequencies or intensity. For example, when PATCH-RND is used blindly on covariates with non-global semantics, the biased model may rely on said semantics; this in turn guides the predictive model to ignore these semantics and, thus, lose predictive performance. Alternatively, when nuisances are global, PATCH-RND may corrupt them. For example in detecting cows and penguins, other nuisance animals (like dogs) may co-occur with cows more often; PATCH-RND would corrupt this nuisance animal. Using PATCH-RND in a B-SCAM for such tasks could lead to non-robust predictive models that rely on corrupted nuisances.

Our experiments suggest that it might be possible to guard against performance degradation due to blind usage of semantic corruptions if the corruption parameter is made a hyperparameter and selected using standard validation schemes. In both classifying waterbirds and NLI, there exist non-global semantics, like small beaks and individual words, that are not corrupted by PATCH-RND and NGRAM-RND respectively. However, in our Waterbirds and NLI experiments, we show models built using semantic corruptions, with validated size choices, close more than 80% of the gap in test performance between ERM and the methods that use known nuisances. Now, imagine the extreme case of running NURD, POE, DFL with a semantic corruption that destroys all information in the covariates. Biased models would predict like random chance, and the resulting predictive models would be no less robust than ERM. On the other hand, methods like JTT perform at least as well as their vanilla versions as long as the validation strategy used in vanilla JTT covers the identity function as a corruption. Future work could consider combining semantic corruptions as a way to better retain of nuisances. Given the validation strategies for B-SCAMs, a practitioner can easily validate over both single and hybrid corruptions.

**Summary.** Semantic corruptions power B-SCAMs to build models robust to spurious correlations using knowledge about the semantic features. Additional knowledge is always required to achieve such robustness, and existing work assumes access to nuisance annotations or that ERM-trained models rely on nuisances. By developing semantic corruptions, we give an approach to use a new kind of additional knowledge, thereby enlarging the set of tasks where one can build robust models. As discussed above, our experiments show that using semantic corruptions in B-SCAMs leads to models more robust than ERM and JTT even when the corruptions may have corrupted some nuisances. These two properties demonstrate the value of semantic corruptions as a way to build robust models.

## Acknowledgements

The authors were supported by NIH/NHLBI Award R01HL148248, NSF Award 1922658 NRT-HDR: FU-TURE Foundations, Translation, and Responsibility for Data Science, NSF CAREER Award 2145542, Grant ONR N00014-23-1-2634, Apple Scholars in AI/ML PhD fellowship, and Samsung Advanced Institute of Technology (Next Generation Deep Learning: From Pattern Recognition to AI). Nitish Joshi is supported by the NSF Graduate Research Fellowship grant number 1839302.

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

# A   Proofs and Discussion on Semantic Corruptions

In this section we give the proofs of Theorem 1 and Proposition 1. The first result shows that even if we know our training and test data are sampled from distributions in a nuisance varying family $\mathcal{F}$, additional assumptions are required in order to learn a predictor that is robust across the entire family.

**Theorem 1.** *For any learning algorithm, there exists a nuisance-varying family $\mathcal{F}$ where predicting with $p_{\perp}(\mathbf{y} = 1 \mid \mathbf{x})$ achieves* 90% *accuracy on all members such that given training data* $\mathbf{y}, \mathbf{x}$ *from one member $p_{tr} \in \mathcal{F}$, the algorithm cannot achieve better accuracy than predicting at random on some $p_{te} \in \mathcal{F}$.*

*Proof.* At a high-level, we setup two nuisance-varying families $\mathcal{F}_1 = \{p_{1,\rho}\}, \mathcal{F}_2 = \{p_{2,\rho}\}$ where

1. There are members of each family that have the same distribution over $(\mathbf{y}, \mathbf{x})$. We let this distribution over $\mathbf{y}, \mathbf{x}$ be the training data.

2. Thus looking at this training data alone, no algorithm can tell which family the test distribution will come from.

3. Then, the proof concludes by showing any predictor that performs better than the chance on all members of $\mathcal{F}_1$, will perform worse than chance on a member of $\mathcal{F}_2$.

**Defining the two families.**   We now define two nuisance-varying families $\mathcal{F}_1 = \{p_{1,\rho}\}$ and $\mathcal{F}_2 = \{p_{2,\rho}\}$. For $a \in \{-1, 1\}$, and $\alpha \in [0, 1]$ let $\mathbf{R}_\alpha(a)$ be a probability distribution obtained by randomly flipping the sign of $a$ with probability $1 - \alpha$:

$$r \sim \mathbf{R}_\alpha(a) \implies \begin{cases} p(r = a) = \alpha \\ p(r = -a) = 1 - \alpha \end{cases} \tag{2}$$

Then, define the family $\{p_{1,\rho}\}$ as the distributions resulting from the following sampling process:

$$\mathbf{y} \sim \mathbf{R}_{0.5}(1)$$
$$\mathbf{z} \sim \mathbf{R}_\rho(\mathbf{y})$$
$$\mathbf{x}^* \sim \mathbf{R}_{0.9}(\mathbf{y})$$
$$\mathbf{x} = [\mathbf{x}^*, \mathbf{z}]$$

The second family $p_{2,\rho}$ follows the same process except that the positions of the semantic feature and nuisance are flipped $\mathbf{x} = [\mathbf{z}, \mathbf{x}^*]$. **Notice that predicting $\mathbf{y}$ from $\mathbf{x}_1$ in $\mathcal{F}_1$ and from $\mathbf{x}_2$ in $\mathcal{F}_2$, achieves** 90% **accuracy.** In both families, by construction, the following properties hold

$$p_{1,\rho}(\mathbf{y}) = p_{2,\rho}(\mathbf{y}) \qquad p_{1,\rho}(\mathbf{z}, \mathbf{y}) = p_{2,\rho}(\mathbf{z}, \mathbf{y}), \qquad p_{1,\rho}(\mathbf{x}^*, \mathbf{y}) = p_{2,\rho}(\mathbf{x}^*, \mathbf{y}), \qquad \mathbf{x}_1 \perp\!\!\!\perp_{p_{\cdot,\rho}} \mathbf{x}_2 \mid \mathbf{y}.$$

If $\rho \neq 0.9$, due to the flipping of the positions of $\mathbf{x}^*, \mathbf{z}$ between $p_{1,\rho}$ and $p_{2,\rho}$,

$$p_{1,\rho}(\mathbf{x}_1 \mid \mathbf{y}) \neq p_{2,\rho}(\mathbf{x}_1 \mid \mathbf{y}) \qquad p_{1,\rho}(\mathbf{x}_2 \mid \mathbf{y}) \neq p_{2,\rho}(\mathbf{x}_2 \mid \mathbf{y}).$$

But when $\rho = 0.9$, the distributions are the same: $p_{\cdot,\rho}(\mathbf{x}_1 \mid \mathbf{y}) \overset{\mathrm{d}}{=} p_{\cdot,\rho}(\mathbf{x}_2 \mid \mathbf{y}) \implies p_{1,0.9}(\mathbf{y}, \mathbf{x}) = p_{2,0.9}(\mathbf{y}, \mathbf{x})$. With this we let the training data come from $p_{tr} = p_{1,0.9}$.

**Reducing accuracy computation to summing conditional probabilities.**   Now, we express the accuracy of any predictor $f(x_1, x_2) \in \{-1, 1\}$ of $p_{1,\rho}$:

$$\begin{aligned} \mathrm{ACC}_f(p_{1,\rho}) &= \mathbb{E}_{p_{1,\rho}(\mathbf{y}, \mathbf{x}_1, \mathbf{x})} \mathbf{1}[\mathbf{y} = f(\mathbf{x}_1, \mathbf{x}_2)] \\ &= \sum_{x_1, x_2} p_{1,\rho}(\mathbf{y} = f(x_1, x_2), \mathbf{x}_1 = x_1, \mathbf{x}_2 = x_2) \\ &= \sum_{x_1, x_2} p_{1,\rho}(\mathbf{x}_1 = x_1, \mathbf{x}_2 = x_2 \mid \mathbf{y} = f(x_1, x_2)) p_{1,\rho}(\mathbf{y} = f(x_1, x_2)) \\ &= 0.5 \sum_{x_1, x_2} p_{1,\rho}(\mathbf{x}_1 = x_1, \mathbf{x}_2 = x_2 \mid \mathbf{y} = f(x_1, x_2)) \end{aligned} \tag{3}$$

With this expression, we have reduced computing the accuracy of a model $f(x_1, x_2)$ to taking one from a pair of numbers — either $p_{1,\rho}(\mathbf{x}_1 = x_1, \mathbf{x}_2 = x_2 \mid \mathbf{y} = 1)$ or $p_{1,\rho}(\mathbf{x}_1 = x_1, \mathbf{x}_2 = x_2 \mid \mathbf{y} = -1)$ based on what $f(x_1, x_2)$ predicts — for each possible value of $x_1, x_1 \in \{-1, 1\}^2$, summing them and multiplying by 0.5.

| $(x_1, x_2)$ | $(-1, -1)$ | $(-1, 1)$ | $(1, -1)$ | $(1, 1)$ | $\text{ACC}_f(p_{1,0})$ acc | $\text{ACC}_f(p_{1,1})$ | min |
|---|---|---|---|---|---|---|---|
| 0 | 1 | 1 | 1 | 1 | 0.50 | 0.50 | 0.50 |
| 1 | 1 | 1 | 1 | -1 | 0.55 | 0.05 | 0.05 |
| 2 | 1 | 1 | -1 | 1 | 0.05 | 0.55 | 0.05 |
| 3 | 1 | 1 | -1 | -1 | 0.10 | 0.10 | 0.10 |
| 4 | 1 | -1 | 1 | 1 | 0.95 | 0.45 | 0.45 |
| 5 | 1 | -1 | 1 | -1 | 1.00 | 0.00 | 0.00 |
| 6 | 1 | -1 | -1 | 1 | 0.50 | 0.50 | 0.50 |
| 7 | 1 | -1 | -1 | -1 | 0.55 | 0.05 | 0.05 |
| 8 | -1 | 1 | 1 | 1 | 0.45 | 0.95 | 0.45 |
| 9 | -1 | 1 | 1 | -1 | 0.50 | 0.50 | 0.50 |
| 10 | -1 | 1 | -1 | 1 | 0.00 | 1.00 | 0.00 |
| 11 | -1 | 1 | -1 | -1 | 0.05 | 0.55 | 0.05 |
| $\Longrightarrow$ 12 | -1 | -1 | 1 | 1 | 0.90 | 0.90 | **0.90** |
| 13 | -1 | -1 | 1 | -1 | 0.95 | 0.45 | 0.45 |
| 14 | -1 | -1 | -1 | 1 | 0.45 | 0.95 | 0.45 |
| 15 | -1 | -1 | -1 | -1 | 0.50 | 0.50 | 0.50 |

**Table 7:** The 16 different functions that are possible when predicting a label in $\{-1, 1\}$ from $\mathbf{x} \in \{-1, 1\}^2$. We compute the accuracies on $p_{1,0}, p_{1,1}$ and report the minimum of the two. The only predictor that achieves better than random chance accuracy (denoted by $\Longrightarrow$) is $f(x_1, x_2) = x_1$.

**Showing only a semantic predictor can achieve better accuracy than random chance on $\mathcal{F}_1$.** Next, we will show that the only way to achieve better accuracy than random chance on every member of $\mathcal{F}_1$ is to predict with $f(x_1, x_2) = x_1$. To show this, we will express the accuracy computation for two distributions $p_{1,0}$ and $p_{1,1}$ by constructing a table of values of $p_{1,\rho}(\mathbf{x}_1 = x_1, \mathbf{x}_2 = x_2 \mid \mathbf{y} = 1)$ and $p_{1,\rho}(\mathbf{x}_1 = x_1, \mathbf{x}_2 = x_2 \mid \mathbf{y} = -1)$ for $\rho = 0$ and $\rho = 1$ separately.

By definition of accuracy from eq. (3), the accuracy of any predictor $f(x_1, x_2)$ comes down to picking one from the pair of numbers — left one if prediction if 1 and right otherwise — from each element in the table, summing them and multiplying by 0.5. There are 16 possible functions (2 possible predictions each for 4 combinations of $x_1, x_2$) and we enumerate them in table 7, showing that only $f^*(x_1, x_2) = x_1$ will perform better than chance on both distributions $p_{1,0}$ and $p_{1,1}$.

**No predictor can achieve better accuracy than random on both $\mathcal{F}_1$ and $\mathcal{F}_2$.** The earlier parts showed that the only predictor that achieves better accuracy than random chance on all of $\mathcal{F}_1$ is one that only relies on $\mathbf{x}_1$, which equals the semantic feature $\mathbf{x}^*$ under $p_{1,\rho}$. However, under $p_{2,\rho}$, $\mathbf{x}_1$ is the nuisance $\mathbf{z}$. Then, the predictor $f^*(x_1, x_2) = x_1$ has zero accuracy under $p_{2,0}$ because under $p_{2,0}$, we have $\mathbf{z} \sim R_0(\mathbf{y})$

which means $\mathbf{z} \neq \mathbf{y}$ with probability one:

$$\text{ACC}_{f^*}(p_{2,0}) = \sum_{x_1,x_2} p_{2,0}(\mathbf{y} = f(x_1, x_2), \mathbf{x}_1 = x_1, \mathbf{x}_2 = x_2) = \sum_{x_1,x_2} p_{2,0}(\mathbf{y} = x_1, \mathbf{z} = x_1, \mathbf{x}_2 = x_2) = 0 \quad (4)$$

$\square$

## A.1 Semantic corruptions, biased models, and proof of proposition 1

We give the definition of a semantic corruption here and discuss how it implies alternative intuitive definitions before presenting the proof of proposition 1 on using corruptions to build biased models.

**Definition 3** (Semantic Corruption). *A semantic corruption is a transformation of the covariates $T(\mathbf{x}, \boldsymbol{\delta})$, where $\boldsymbol{\delta}$ is a random variable such that $\boldsymbol{\delta} \perp\!\!\!\perp (\mathbf{y}, \mathbf{z}, \mathbf{x}, \mathbf{x}^*)$, if*

$$\forall p_D \in \mathcal{F} \quad T(\mathbf{x}, \boldsymbol{\delta}) \perp\!\!\!\perp_{p_D} \mathbf{x}^* \mid \mathbf{z}.$$

Two other plausible definitions that come to mind are $T(\mathbf{x}, \boldsymbol{\delta}) \perp\!\!\!\perp_{p_\perp} \mathbf{x}^*$ and that $\mathbf{y} \perp\!\!\!\perp_{p_D} T(\mathbf{x}, \boldsymbol{\delta}) \mid \mathbf{z}$. These are both intuitive properties that can be asked of a semantic corruption that is supposed to discards all information about semantics, provided that the $\mathbf{z}$ which we wish to retain holds no information on it (which is the case under $p_\perp$). We now show that definition 3 implies these two.

From the definition that if $T(\mathbf{x}, \boldsymbol{\delta})$ is a semantic corruption, then it also holds that $T(\mathbf{x}, \boldsymbol{\delta}) \perp\!\!\!\perp_{p_\perp} \mathbf{x}^*$: since $\mathbf{x}^* \perp\!\!\!\perp_{p_\perp} \mathbf{z}$

$$p_\perp(T(\mathbf{x}, \boldsymbol{\delta}), \mathbf{x}^*) = \mathbb{E}_{p_\perp(\mathbf{z})} p_\perp(T(\mathbf{x}, \boldsymbol{\delta}), \mathbf{x}^* \mid \mathbf{z}) = \mathbb{E}_{p_\perp(\mathbf{z})} p_\perp(T(\mathbf{x}, \boldsymbol{\delta}) \mid \mathbf{z}) p_\perp(\mathbf{x}^* \mid \mathbf{z}) \quad (5)$$

$$= p_\perp(\mathbf{x}^*) \mathbb{E}_{p_\perp(\mathbf{z})} p_\perp(T(\mathbf{x}, \boldsymbol{\delta}) \mid \mathbf{z}) = p_\perp(\mathbf{x}^*) p_\perp(T(\mathbf{x}, \boldsymbol{\delta})). \quad (6)$$

A semantic corruption satisfies the second definition also because

$$p_D(\mathbf{y}|T(\mathbf{x}), \mathbf{z}) = \int p_D(\mathbf{y}|\mathbf{x}^*, T(\mathbf{x}), \mathbf{z}) p_D(\mathbf{x}^*|\mathbf{z}, T(\mathbf{x})) d\mathbf{x}^* = \int p_D(\mathbf{y}|\mathbf{x}^*, \mathbf{z}) p_D(\mathbf{x}^*|\mathbf{z}, T(\mathbf{x})) d\mathbf{x}^*$$

$$= \int p_D(\mathbf{y}|\mathbf{x}^*, \mathbf{z}) p_D(\mathbf{x}^*|\mathbf{z}) d\mathbf{x}^* = p_D(\mathbf{y}|\mathbf{z}) \quad (7)$$

First transition adds in integration over the values of $\mathbf{x}^*$, second one uses the property of the nuisance varying family that $\mathbf{x} \perp\!\!\!\perp_{p_D} \mathbf{y}|\mathbf{z}, \mathbf{x}^*$ and therefore it is also conditionally independent for any $T(\mathbf{x}, \boldsymbol{\delta})$. Then the third transition is due to $T(\mathbf{x}, \boldsymbol{\delta})$ being a semantic corruption. The next result shows that the more our semantic corruption captures information about the nuisance that is relevant to predicting $\mathbf{y}$, the better we can approximate learning under $p_\perp$, which would yield the optimal risk-invariant predictor over $\mathcal{F}$ (Makar et al., 2022).

### A.1.1 Proof of proposition 1.

Now, using the property in eq. (7) that holds for semantic corruptions, we prove proposition 1.

**Proposition 1.** *Let $T : \mathbf{X} \times \mathbf{R}^d \to \mathbf{X}$ be a function. Assume the r.v. $p_{tr}(\mathbf{y} \mid T(\mathbf{x}, \boldsymbol{\delta}))^{-1}$ has a bounded second moment under the distribution $p_\perp(\mathbf{y}, \mathbf{z}, \mathbf{x})p(\boldsymbol{\delta})$, and that $p_{tr}(\mathbf{y} \mid T(\mathbf{x}, \boldsymbol{\delta}))$ and $p_{tr}(\mathbf{y} \mid \mathbf{z})$ satisfy*

$$\mathbb{E}_{p_\perp(\mathbf{y}, \mathbf{z}, \mathbf{x})p(\boldsymbol{\delta})} p_{tr}(\mathbf{y} \mid T(\mathbf{x}, \boldsymbol{\delta}))^{-2} \leq m^2, \qquad \mathbb{E}_{p_\perp(\mathbf{y}, \mathbf{z}, \mathbf{x})p(\boldsymbol{\delta})} |p_{tr}(\mathbf{y} \mid T(\mathbf{x}, \boldsymbol{\delta})) - p_{tr}(\mathbf{y} \mid \mathbf{z})|^2 = \epsilon^2.$$

*Then, the $L_1$ distance between $p_\perp(\mathbf{y}, \mathbf{x})$ and $p_T(\mathbf{y}, \mathbf{x})$ is bounded: $\|p_\perp(\mathbf{y}, \mathbf{x}) - p_T(\mathbf{y}, \mathbf{x})\|_1 \leq m\epsilon$. For a semantic corruption that also satisfies $\mathbf{y} \perp\!\!\!\perp_{p_{tr}} \mathbf{z} \mid T(\mathbf{x}, \boldsymbol{\delta})$ the inequalities hold with $\epsilon = 0$.*

*Proof.* The $L_1$ distance between the distributions is bounded from above by a $p_\perp$-weighted $L_1$ distance between $p_{tr}(\mathbf{y} \mid \mathbf{z})$ and $p_{tr}(\mathbf{y} \mid T(\mathbf{x}))$, upto a constant:

$$\int_{y,x} |p_\perp(\mathbf{y}, \mathbf{x}) - p_T(\mathbf{y}, \mathbf{x}))| \, dy dx \tag{8}$$

$$= \int_{y,x} \left| \int_z p_{tr}(\mathbf{y}) p_{tr}(\mathbf{y}, \mathbf{z}, \mathbf{x}) p(\boldsymbol{\delta}) \left[ \frac{1}{p_{tr}(\mathbf{y} \mid \mathbf{z})} - \frac{1}{p_{tr}(\mathbf{y} \mid T(\mathbf{x}, \boldsymbol{\delta}))} \right] dz \right| dy dx \tag{9}$$

$$= \int_{y,x} \left| \int_z p_{tr}(\mathbf{y}) p_{tr}(\mathbf{y}, \mathbf{z}, \mathbf{x}) p(\boldsymbol{\delta}) \left[ \frac{p_{tr}(\mathbf{y} \mid T(\mathbf{x})) - p_{tr}(\mathbf{y} \mid \mathbf{z})}{p_{tr}(\mathbf{y} \mid \mathbf{z}) p_{tr}(\mathbf{y} \mid T(\mathbf{x}, \boldsymbol{\delta}))} - \right] dz \right| dy dx \tag{10}$$

$$= \int_{y,x} \left| \mathbb{E}_{p_{tr}(\mathbf{z}) p(\boldsymbol{\delta})} \frac{p_{tr}(\mathbf{y})}{p_{tr}(\mathbf{y} \mid T(\mathbf{x}, \boldsymbol{\delta}))} p(\mathbf{x} \mid \mathbf{y}, \mathbf{z}) \left[ p_{tr}(\mathbf{y} \mid T(\mathbf{x}, \boldsymbol{\delta})) - p_{tr}(\mathbf{y} \mid \mathbf{z}) \right] \right| dy dx \tag{11}$$

$$\leq \int_{y,x} \mathbb{E}_{p_{tr}(\mathbf{z}) p(\boldsymbol{\delta})} \left| \frac{p_{tr}(\mathbf{y})}{p_{tr}(\mathbf{y} \mid T(\mathbf{x}, \boldsymbol{\delta}))} p(\mathbf{x} \mid \mathbf{y}, \mathbf{z}) \left[ p_{tr}(\mathbf{y} \mid T(\mathbf{x}, \boldsymbol{\delta})) - p_{tr}(\mathbf{y} \mid \mathbf{z}) \right] \right| dy dx \tag{12}$$

$$= \int_{y,x,z} p_{tr}(\mathbf{z}) p_{tr}(\mathbf{y}) p(\boldsymbol{\delta}) p(\mathbf{x} \mid \mathbf{y}, \mathbf{z}) \frac{1}{p_{tr}(\mathbf{y} \mid T(\mathbf{x}, \boldsymbol{\delta}))} |p_{tr}(\mathbf{y} \mid T(\mathbf{x}, \boldsymbol{\delta})) - p_{tr}(\mathbf{y} \mid \mathbf{z})| \, dy dx dz \tag{13}$$

$$= \mathbb{E}_{p_\perp(\mathbf{y}, \mathbf{z}, \mathbf{x}) p(\boldsymbol{\delta})} \frac{1}{p_{tr}(\mathbf{y} \mid T(\mathbf{x}, \boldsymbol{\delta}))} |p_{tr}(\mathbf{y} \mid T(\mathbf{x}, \boldsymbol{\delta})) - p_{tr}(\mathbf{y} \mid \mathbf{z})| \tag{14}$$

$$\leq \left( \sqrt{\mathbb{E}_{p_\perp(\mathbf{y}, \mathbf{x}) p(\boldsymbol{\delta})} \frac{1}{p_{tr}(\mathbf{y} \mid T(\mathbf{x}, \boldsymbol{\delta}))^2}} \right) \sqrt{\mathbb{E}_{p_\perp(\mathbf{y}, \mathbf{z}, \mathbf{x}) p(\boldsymbol{\delta})} |p_{tr}(\mathbf{y} \mid T(\mathbf{x}, \boldsymbol{\delta})) - p_{tr}(\mathbf{y} \mid \mathbf{z})|^2} \tag{15}$$

Substituting the bounds from the theorem statement completes the proof of the bound.

Finally, if $T$ is a semantic corruption, by eq. (7), it holds that

$$p_{tr}(\mathbf{y} \mid T(\mathbf{x}, \boldsymbol{\delta}), \mathbf{z}) = p_{tr}(\mathbf{y} \mid \mathbf{z}).$$

Then, if it also holds that $\mathbf{y} \perp\!\!\!\perp_{p_{tr}} \mathbf{z} \mid T(\mathbf{x}, \boldsymbol{\delta})$, it holds that

$$p_{tr}(\mathbf{y} \mid T(\mathbf{x}, \boldsymbol{\delta}), \mathbf{z}) = p_{tr}(\mathbf{y} \mid T(\mathbf{x}, \boldsymbol{\delta})).$$

Together this implies that almost everywhere in $p_{tr}(\mathbf{y}, \mathbf{z}, \mathbf{x}) p(\boldsymbol{\delta})$

$$p_{tr}(\mathbf{y} \mid T(\mathbf{x}, \boldsymbol{\delta})) = p_{tr}(\mathbf{y} \mid \mathbf{z}) \implies \mathbb{E}_{p_\perp(\mathbf{y}, \mathbf{z}, \mathbf{x}) p(\boldsymbol{\delta})} |p_{tr}(\mathbf{y} \mid T(\mathbf{x}, \boldsymbol{\delta})) - p_{tr}(\mathbf{y} \mid \mathbf{z})|^2 = 0.$$

This shows that for a semantic corruption such that $\mathbf{y} \perp\!\!\!\perp_{p_{tr}} \mathbf{z} \mid T(\mathbf{x}, \boldsymbol{\delta})$, it holds that $\epsilon = 0$. $\qquad\square$

# B  Further details about biased-model-based spurious-correlation avoiding methods and related work

**Nurd.** Focusing on mitigating spurious correlations, Puli et al. (2022) identify a conditional that has performance guarantees on every test distribution within a family of distributions with varying nuisance-label relationships: $p_{te} \in \mathcal{F}$. They develop NURD to learn the conditional using data only from $p_{tr} \neq p_{te}$. NURD uses 1) the *nuisance-randomized distribution*, $p_\perp(\mathbf{y}, \mathbf{z}, \mathbf{x}) = p(\mathbf{y}) p_\perp(\mathbf{z}) p(\mathbf{x} \mid \mathbf{y}, \mathbf{z})$, where $\mathbf{z} \perp\!\!\!\perp_{p_\perp} \mathbf{y}$, and 2) an *uncorrelating representation* $r(\mathbf{x})$ for which $\mathbf{z} \perp\!\!\!\perp_{p_\perp} \mathbf{y} \mid r(\mathbf{x})$. NURD builds models of the form $p_\perp(\mathbf{y} \mid r(\mathbf{x}))$ using $r(\mathbf{x})$ that are most informative of the label.

We run reweighting-NURD, which uses a biased model $p_{tr}(\mathbf{y} \mid \mathbf{z})$ as an importance weight to compute loss under the nuisance-randomized distribution: $p_\perp(\mathbf{y}, \mathbf{z}, \mathbf{x}) = \frac{p_{tr}(\mathbf{y})}{p_{tr}(\mathbf{y} \mid \mathbf{z})} p_{tr}(\mathbf{y}, \mathbf{z}, \mathbf{x})$.

To run reweighting-NURD with semantic corruptions, we replace $p_{tr}(\mathbf{y} \mid \mathbf{z})$ with $p_{tr}(\mathbf{y} \mid T(\mathbf{x}))$ for a semantic corruption $T(\mathbf{x})$. Semantic corruptions are noisy functions of $\mathbf{x}$: with noise $\boldsymbol{\epsilon}$ such that $(\mathbf{y}, \mathbf{z}, \mathbf{x}) \perp\!\!\!\perp_{p_D} \boldsymbol{\epsilon}$, $T(\mathbf{x}) = U(\mathbf{x}, \boldsymbol{\epsilon})$. This implies

$$\mathbf{y} \perp\!\!\!\perp_{p_\perp} \boldsymbol{\epsilon} \mid \mathbf{x} \implies \mathbf{y} \perp\!\!\!\perp_{p_\perp} \mathbf{x}, \boldsymbol{\epsilon} \mid \mathbf{x} \implies \mathbf{y} \perp\!\!\!\perp_{p_\perp} T(\mathbf{x}) \mid \mathbf{x}$$

Thus, $r(\mathbf{x}) = \mathbf{x}$ is uncorrelating and $p_\perp(\mathbf{y} \mid \mathbf{x})$ achieves the optimality guarantees in Puli et al. (2022). These optimality guarantees imply that regardless of the test nuisance-label relationship, $p_\perp(\mathbf{y} \mid \mathbf{x})$ will achieve optimal performance within the class of models like $p_\perp(\mathbf{y} \mid r(\mathbf{x}))$.

**End-to-end bias mitigation.** Mahabadi et al. (2019) consider two methods to train a *biased* model $p_{tr}(\mathbf{y} \mid \mathbf{z})$ and a base predictive model jointly to make the base model predict without relying on the biases. The methods use and fine-tune a BERT model (Devlin et al., 2019) and do not propagate the gradients of the biased model to update the common parameters (token embeddings in this case). They propose 1) POE, where the `log` of the product of the predictions (the output probabilities) of the two models is used to compute the classification loss and 2) DFL, where the biased model is used to weight the cross-entropy loss for the base model.

The intuition for POE is that the samples for which the biased model classifies correctly will not contribute to the gradients of the base model; thus the base model focuses more on classifying samples that the biased model misclassifies. The DFL algorithm weights each sample as the biased model's predicted probability of all but the label, exponentiated with $\gamma > 0$. This downweights samples that the biased model classifies correctly which in turn mitigates the base model's reliance on a nuisance which only helps predict the downweighted samples correctly.

Formally, with a biased model $f_\theta(\mathbf{z})$ and a predictive model $f_\gamma(\mathbf{x})$ that output a vector of logits over classes, $\sigma$ denoting the soft-max function that maps logits to class-probabilities, and $\sigma(\cdot)_y$ denoting the softmax-probability of label $y$

$$\text{POE} \quad \max_{\theta,\gamma} \sum_{i \in \texttt{training data}} \log \sigma(f_\theta(\mathbf{z}_i))_{y_i} + \log \sigma(f_\gamma(\mathbf{x}_i))_{y_i} \tag{16}$$

$$\text{DFL} \quad \max_{\theta,\gamma} \sum_{i \in \texttt{training data}} (1 - \sigma(f_\theta(\mathbf{z}_i))_{y_i})^\gamma \log \sigma(f_\gamma(\mathbf{x}_i))_{y_i} \tag{17}$$

Mahabadi et al. (2019) build the biased model $f_\theta$ using known nuisances $\mathbf{z}$. We build this model from a semantic corruption $T(\mathbf{x})$.

**Just Train Twice (JTT).** JTT works in two stages: 1) build an "identification" model via ERM on the training data to isolate samples that are misclassified due to reliance on the nuisances and 2) train a model via ERM on data with the loss for the misclassified samples upweighted (by constant $\lambda$). The identification model in JTT is built to be a biased model. When the identification model equals $p_{tr}(\mathbf{y} \mid \mathbf{z})$, it exactly misclassifies the samples in the groups in the minority group[1]. Upweighting these samples produces a dataset with lesser dependence between the nuisance and the label. Models learned on the upweighted data depend more on the semantics. See algorithm 1 for pseudocode.

---

**Algorithm 1** JTT.

**Input:** Training set $D$ and hyperparameters $T$ and $\lambda_{\text{up}}$. **Stage one: identification**
1. Train identification model $f_\theta$ on $D$ via ERM for $T$ steps.
2. Construct the errors set of training examples misclassified by $f_\theta$.
**Stage two: upweighting identified points**
3. Construct upsampled dataset $D_{\text{up}}$ containing examples in the error set repeated $\lambda_{\text{up}}$ times and all other examples once.
4. Train final model $f_\gamma$ on $D_{\text{up}}$ via ERM.

---

In this work, we build the identification model on semantic corruptions i.e. we learn $f_\theta$ to predict $\mathbf{y}$ from $T(\mathbf{x})$. The training samples to be upweighted are the ones misclassified when predicting with the identification model on semantic-corrupted versions of the sample, i.e. $T(\mathbf{x})$. The second stage is run as in (Liu et al., 2021) with training data.

**Optimization-generalization Dilemma** Like many other algorithms in the OOD generalization literature, training B-SCAMss based on semantic corruptions may also suffer from obstacles due to optimization and generalization: employing statistical constraints to handle distribution shift may not build models that perform well OOD due to overfitting (Wald et al., 2022), training difficulties (Chen et al., 2022; Zhang et al.,

---

[1] The minority group is the set of samples that the nuisance misclassifies. For example, when $p_{tr}(\mathbf{y} = \mathbf{z}) > p_{tr}(\mathbf{y} \neq \mathbf{z})$, then the minority group is the set of samples with $\mathbf{y} \neq \mathbf{z}$ because using only the nuisance results in predicting $\mathbf{y} = b$ where $\mathbf{z} = b$.

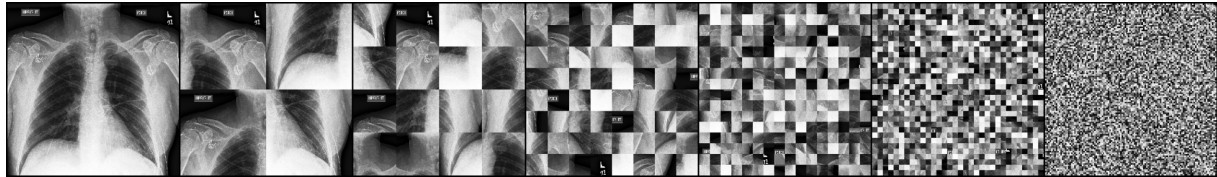

**Figure 3:** Example of PATCH-RND of a chest X-ray image. The image is followed by PATCH-RNDs of size $112, 56, 28, 14, 7, 2$.

2022; Chen et al., 2024), or reliance on inappropriate inductive biases (Nagarajan et al., 2020; Puli et al., 2023). Some approaches in the literature can alleviate these difficulties: two-stage methods incorporate the OOD objective only when training smaller models on top of large ones (Chen et al., 2022; Zhang et al., 2022; Chen et al., 2024; Yong et al., 2023; Kirichenko et al., 2022), subsampling instead of weighting (Sagawa et al., 2020; Idrissi et al., 2022), or large $\ell_2$ regularization (Sagawa et al., 2019).

In our implementations we use validation data and regularization to tune parameters for the weighted-ERM algorithm as proposed in the original papers of the B-SCAMs we experiment with. As ERM is standard practice, there are no new optimization difficulties but generalization difficulties can occur due to overfitting (Wald et al., 2022; Puli et al., 2023). Any improvements in generalization in weighted-ERM will lead to improvements in models built by B-SCAMs with biased models from semantic corruptions.

## C Further experimental details

### C.1 Remark on baseline corruptions

NURD with the baseline corruption GAUSS-NOISE outperforms ERM and closes $80\%$ of the gap between ERM and known-$\mathbf{z}$ NURD in table 2. We explain such an improvement as a consequence of GAUSS-NOISE corrupting semantics more than it corrupts nuisances; we explain below. In tasks like waterbirds, nuisances are present in most if not all patches of the image regardless of where the patches appear. On the other hand, semantic features are localized to a few adjacent patches (like the birds parts appearing next to each other). When nuisances are present is many more patches than the semantics, adding gaussian noise to all pixels corrupts semantics more than nuisances. To see why, consider meauusurements of a quantity as a gaussian random variable with the quantity as its mean. More measurements lead to better estimates of the mean.

### C.2 Implementation details

Each experiment in the paper was run on up to 2 RTX8000 GPUs. The hyperparameters for methods that use known nuisances in the training data, like NURD, POE, DFL are tuned on validation data from the training distribution. For NURD, we select corruption hyperparameters using the mean of the balanced validation accuracy across 10 seeds. We do the same when using semantic corruptions.

**Experimental details for Waterbirds** For the NURD setup, the training, validation, and test datasets have $3020, 756, 800$ samples respectively. We use a single architecture to parameterize the predictive model and the weight model in this experiment: two fully connected layers on top of a ResNet18 initialized at weights pretrained on Imagenet. We use the same training procedure for NURD with known nuisances or with semantic corruptions. Both models are trained with cross-entropy. The weight model is optimized with the default Adam optimizer for 20 epochs with a batch size of 64. The predictive model is optimized with the Adam optimizer for 20 epochs with a learning rate of 0.0002, a weight decay of 0.01, and a batch size of 250.

For the JTT setup, the training, validation, and test datasets have $4795, 1199, 5794$ samples respectively. For JTT, we use the same model and model parameters as Liu et al. (2021) using their released code. We repeat the details here for completeness. The model for both stages of JTT is a ResNet-50. Both models are optimized by stochastic gradient descent (SGD) with momentum 0.9, weight decay 1.0, and learning rate $1 \times 10^{-5}$. Both models are trained for 300 epochs with batch size 64, using batch normalization and no data augmentation. The identification model used to select samples to upweight corresponds to epoch 60 and the upweighting constant is $\lambda = 100$.

**Experimental details for cardiomegaly detection.** The training, validation, and test datasets are fixed across seeds and have $18000, 2000, 1000$ samples respectively. To run reweighting-NURD, we use a single architecture to parameterize the predictive model and the weight model in this experiment: two fully connected layers on top of a ResNet18 initialized at weights pretrained on Imagenet. In known-nuisance NURD with the hospital as the nuisance, the biased model is an estimate of $p_{tr}(\mathbf{y} \mid \text{hospital})$, which is obtained by binning the samples based on the hospital and averaging the labels. We use the same training procedure for NURD with known nuisances or with semantic corruptions. Both weight and predictive models are trained with cross-entropy. The weight model and the predictive model are optimized with the Adam optimizer over 25 epochs with a batch size of 256, and learning rate 0.001.

**Implementation details for nli** For POE and DFL, we build classifiers by fine-tuning a pretrained BERT model (Devlin et al., 2019) on the data. We follow the same training procedure and hyperparameter details as used in Mahabadi et al. (2019) — models were trained on the MNLI training dataset which consists of 392k examples, with a learning rate of $2 \times 10^{-5}$ with a batch size of 8 using the Adam Optimizer. All models are trained for 3 epochs. The development set contains 9815 examples and the HANS test contains 30000 examples. Since the HANS dataset has only two labels — 'entailment' and 'non-entailment' — we combine the neutral and contradiction classes during inference on HANS.

For the JTT setup, Liu et al. (2021) mix the training and development sets from MNLI and create their own training, validation, and test sets of sizes $206175, 82462, 123712$ respectively. For JTT, we use the same model and model parameters as Liu et al. (2021) using their released code. We use the optimal hyperparameters reported in Liu et al. (2021) for the learning rate, weight decay, and the upweighting constant. We repeat the details here for completeness. The model for both stages of JTT is a pretrained BERT model that is finetuned during training. Both models are optimized by the AdamW optimizer with clipping for the predictive model, no weight decay, and an initial learning rate of $2 \times 10^{-5}$. Both models are trained for 5 epochs with batch size 32 and dropout. The identification model used to select samples to upweight corresponds to epoch 2 for vanilla JTT (reported optimal in Liu et al. (2021)); for JTT with semantic corruption, we select one from $2, 3$ using validation group annotations. For both, the upweighting constant is $\lambda = 6$. Our runs with these parameters did not yield the test worst-group accuracy reported in (Liu et al., 2021) (72.6%); our experiments yielded a test worst-group accuracy 71.3%. We expect this may be due to the differences in the random seed; JTT is sensitive to hyperparameters and differences in order of batches may result in drops in performance.

In NGRAM-RND, when the number of words in the sentence is not a multiple of $n$, there will be one $k$-gram ($k < n$). In implementing NGRAM-RND, we ensure that the position of this k-gram is randomized i.e. we make sure that it does not always occur at the end of the sentence, for example. NGRAM-RND is implemented before word-piece tokenization (which BERT uses), to ensure that we randomize words instead of subwords. We also create a small HANS-like development set, which is used to tune the size parameter. This set is constructed by randomly sampling 1000 examples from the HANS training set, which has zero overlap with the HANS test set.

### C.3 Full results tables and additional experiments

We give the results for all size parameters; see table 10, table 11, table 12, table 13, and table 14. To report the same metrics as in Mahabadi et al. (2019) for POE and DFL and Puli et al. (2022) for NURD, we report standard error for NURD and standard deviation for POE and DFL .

#### C.3.1 Results on Adversarial NLI (Nie et al., 2019) and CAD (Kaushik et al., 2019)

In table 15 and table 16, we report evaluations of POE and DFL models on the adversarial ANLI (Nie et al., 2019) and the counterfactually augmented dataset (Kaushik et al., 2019).

#### C.3.2 Additional experiments

**Experiments with weaker spurious correlations.** To verify the effectiveness of the semantic corruptions for powering B-SCAMs like JTT that rely on assumptions on ERM-trained models, we experiment with a modified version of the Waterbirds dataset. In the modified dataset, the spurious feature predicts the label

only 75% of the time; this is weaker than the 93% in the original dataset and the invariant relationship which achieves $> 85\%$ accuracy across all groups. We ran ERM, JTT, and corruption-powered JTT. For both versions of JTT, we tune over the same hyperparameters as in Liu et al. (2021).

The results in table 8 show that corruption-powered JTT is better than vanilla JTT and ERM. The improvement of corruption-powered JTT over vanilla JTT increases from 0.5% in table 3 to 4.4% in table 8; this indicates that vanilla JTT is more sensitive to the strength of the spurious correlation than corruption-powered JTT.

**Table 8:** Test worst-group (WG) accuracies of JTT on modified waterbirds where the spurious correlation is weaker than the invariant relationship. Corruption-powered JTT outperforms ERM, vanilla JTT, and JTT with baseline corruptions (RAND-CROP, GAUSS-NOISE) by $\geq 4.4\%$.

| Method | test WG acc. |
|---|---|
| *Vanilla* JTT | 78.6% |
| PATCH-RND | 84.6% |
| ROI-MASK | 85.2% |
| FREQ-FILT | 83.2% |
| INT-FILT | 83.0% |
| RAND-CROP | 76.2% |
| GAUSS-NOISE | 75.9% |
| ERM | 76.1% |

**Experiments with multiple spurious features.** We run ROI-MASK-powered NURD with a modified version of the ColorFulMNIST dataset (Yong et al., 2023). The images consist of $42 \times 42 \times 3$ pixels, with the middle $14 \times 14$ forming the MNIST image showing a 0 or a 1 and the rest being background patches. The digit in the middle predicts the binary label 1 or 0 with 75% accuracy. Given some $p \in [0,1]$, this dataset sets each of the background patch colors deterministically based on the image in the middle with probability $p$; with probability $1 - p$, each background is a random color (see figure 5 in (Yong et al., 2023).) We generate the training data with $p = 0.9$, and the validation and test data with $p = 0$.

ROI-MASK-powered NURD with central-ROI sizes 14 and 28 achieves test accuracies 71.1% and 70.3% respectively, beating ERM which achieves 51.7% because it relies more on the background colors. PATCH-RND is not suited for this experiment because the different nuisance colors are chosen based on the patch position, and PATCH-RND randomizes patch positions which corrupt these nuisances.

**Experiments showing that corrupting the semantics is the reason behind the improved ood performance in corruption-powered b-scams.** First, we show that corruptions actually do corrupt semantics, taking PATCH-RND as the example. We focus on the Waterbirds dataset to show how patch size affects semantics. For this investigation, we construct training and test datasets where the label and nuisance are independent and build models for predicting the label.

The results are in table 9 and show that as patch-size decreases, more semantic information is lost. These results mean that for patch sizes $< 28$, a biased model built from the corrupted image cannot predict the label well using semantics alone; the accuracy of random chance is 50%. As the label is independent of the nuisance, a lower accuracy means more semantic information is corrupted. However, on the original dataset, our biased models at these patch sizes achieve at least 85% accuracy in predicting the label from the corrupted images, meaning that they rely mostly on the nuisance.

**Table 9:** Accuracy of predicting the label from the image corrupted by PATCH-RND as patch-size decreases. As the label is independent of the nuisance, a lower accuracy means that more semantic information is corrupted.

| PATCH-RND size | Accuracy |
|---|---|
| Full image | 86% |
| 112 | 76% |
| 56 | 73% |
| 28 | 64% |
| 14 | 58% |
| 7 | 57% |

Second, to show that corruptions actually do help, we ran the full NURD algorithm on the Waterbirds dataset from (Puli et al., 2022) with a biased model built directly on the uncorrupted covariates; that is we train a model with ERM to predict **y** from **x** and use it as the biased model in NURD. The resulting test accuracy is $< 70\%$. When using patch-sizes under 28, the PATCH-RND-powered NURD algorithm achieves a test accuracy of nearly 87%. This shows that the corruption of semantics is directly responsible for improving model robustness.

**Table 10:** Average accuracies and standard deviation over 4 seeds of POE and DFL with semantic corruptions on the HANS dataset. The results for *known* POE and DFL from Mahabadi et al. (2019), where both methods use known nuisances. For both methods, selecting the size hyperparameter based on the average accuracy on a small dataset (1000 samples) from the test distribution gives $n = 3$. With this size, POE with NGRAM-RND performs better than known-nuisance POE while DFL with NGRAM-RND closes 84% of the gap between ERM and known-nuisance DFL .

| z | POE | DFL |
|---|---|---|
| *Known* | $66.3 \pm 0.6\%$ | $69.3 \pm 0.2\%$ |
| 1-gram | $65.7 \pm 2.0\%$ | $66.5 \pm 1.5\%$ |
| 2-gram | $66.0 \pm 0.9\%$ | $68.5 \pm 0.7\%$ |
| 3-gram | $66.7 \pm 1.5\%$ | $68.4 \pm 1.5\%$ |
| 4-gram | $66.2 \pm 2.9\%$ | $65.0 \pm 2.0\%$ |
| ERM | — | $63.6\%$. |

**Table 11:** Mean and standard error of test accuracy across 10 seeds of NURD on classifying waterbirds. *Known*-nuisance NURD uses a label for the type of background as the nuisance. Selecting the size hyperparameter based on the average accuracy over 10 seeds on the validation dataset gives 14 for PATCH-RND, 196 for ROI-MASK, 168 for FREQ-FILT, and 0.2 for INT-FILT. Consider the gap between ERM and known-nuisance NURD. NURD with PATCH-RND, ROI-MASK, FREQ-FILT, and INT-FILT close $99\%, 99\%, 82\%, 99\%$ of the gap respectively. NURD with these semantic corruptions outperforms ERM and NURD with RAND-CROP and GAUSS-NOISE. NURD with all semantic corruptions outperforms ERM (69.2%).

| | *known* z | RM 196 | RM 168 | RM 140 | RM 112 | PR 7 | PR 14 | PR 28 | PR 56 | ERM |
|---|---|---|---|---|---|---|---|---|---|---|
| Mean | 87.2% | 86.9% | 86.6% | 86.2% | 86.3% | 85.6% | 86.9% | 82.5% | 84.9% | 68.0% |
| Std. err. | 1.0% | 1.1% | 1.2% | 1.8% | 1.6% | 1.4% | 1.2% | 2.0% | 1.4% | 1.9% |

| | | FF 196 | FF 168 | FF 140 | FF 112 | IF 0.1 | IF 0.2 | IF 0.3 | IF 0.4 | |
|---|---|---|---|---|---|---|---|---|---|---|
| Mean | | 83.8% | 83.5% | 81.0% | 80.3% | 81.2% | 86.9% | 85.0% | 81.9% | |
| Std. err. | | 1.2% | 1.1% | 1.4% | 1.7% | 1.7% | 1.1% | 1.5% | 1.7% | |

| | | RAND-CROP | | | | GAUSS 0.01 | GAUSS 0.25 | GAUSS 1 | GAUSS 4 | |
|---|---|---|---|---|---|---|---|---|---|---|
| Mean | | 73.7% | | | | 75.8% | 74.1% | 78.0% | 83.9% | |
| Std. err. | | 2.0% | | | | 3.2% | 3.1% | 3.4% | 1.4% | |

**Table 12:** Mean and standard error of test accuracy across 10 seeds of NURD on detecting cardiomegaly from chest X-rays. *Known*-nuisance NURD uses the hospital as the nuisance. Selecting the corruption parameters based on the mean accuracy over 10 seeds on the validation dataset gives 14 for PATCH-RND, 196 for ROI-MASK, 168 for FREQ-FILT, and 0.1 for the INT-FILT. Consider the gap between ERM and known-nuisance NURD. NURD with PATCH-RND, ROI-MASK, FREQ-FILT, and INT-FILT close $72\%, 82\%, 65\%, 35\%$ of the gap respectively. NURD with semantic corruptions outperforms NURD with baseline augmentations RAND-CROP and GAUSS-NOISE. NURD with PATCH-RND and ROI-MASK outperforms ERM for all size parameters.

| | *known* z | RM 196 | RM 168 | RM 140 | RM 112 | PR 7 | PR 14 | PR 28 | PR 56 | ERM |
|---|---|---|---|---|---|---|---|---|---|---|
| Mean | 81.7% | 78.7% | 78.3% | 77.2% | 73.6% | 76.2% | 77.0% | 74.9% | 74.3% | 65.3% |
| Std. err. | 0.3% | 0.3% | 0.8% | 0.8% | 0.7% | 1.2% | 1.2% | 1.0% | 1.4% | 1.1% |

| | FF 196 | FF 168 | FF 140 | FF 112 | IF 0.1 | IF 0.2 | IF 0.3 | IF 0.4 |
|---|---|---|---|---|---|---|---|---|
| Mean | 74.4% | 76.0% | 75.3% | 71.3% | 71.0% | 68.0% | 62.0% | 57.1% |
| Std. err. | 1.5% | 0.6% | 0.9% | 1.6% | 1.0% | 1.6% | 1.8% | 3.2% |

| | RAND-CROP | | | | GAUSS 0.01 | GAUSS 0.25 | GAUSS 1 | GAUSS 4 |
|---|---|---|---|---|---|---|---|---|
| Mean | 59.9% | | | | 62.3% | 63.5% | 68.0% | 69.0% |
| Std. err. | 2.1% | | | | 3.7% | 3.4% | 1.1% | 1.9% |

**Table 13:** Test worst-group accuracies of JTT with semantic corruptions on waterbirds. Selecting the corruption hyperparameters on the validation worst-group accuracy gives size 14 for PATCH-RND, size 196 for ROI-MASK, size 112 for FREQ-FILT, and threshold 0.4 for INT-FILT. JTT with these semantic corruptions outperforms ERM, vanilla JTT, and JTT with the baseline corruptions RAND-CROP and GAUSS-NOISE. JTT with PATCH-RND and ROI-MASK outperforms JTT with the baseline corruptions and ERM for all sizes.

| *Vanilla* JTT | RM 196 | RM 168 | RM 140 | RM 112 | PR 7 | PR 14 | PR 28 | PR 56 | ERM |
|---|---|---|---|---|---|---|---|---|---|
| 86.5% | 88.2% | 88.0% | 86.9% | 86.2% | 89.3% | 89.0% | 88.9% | 89.1% | 72% |

| FF 196 | FF 168 | FF 140 | FF 112 | IF 0.1 | IF 0.2 | IF 0.3 | IF 0.4 |
|---|---|---|---|---|---|---|---|
| 82.5% | 84.5% | 85.2% | 87.2% | 69.1% | 80.0% | 81.7% | 87.0% |

| RAND-CROP | | | | GAUSS 0.01 | GAUSS 0.25 | GAUSS 1 | GAUSS 4 |
|---|---|---|---|---|---|---|---|
| 75% | | | | 0.0% | 0.0% | 71.0% | 0.0% |

**Table 14:** Worst-group and average test accuracies of JTT with semantic corruptions on NLI. JTT with PREM-MASK and NGRAM-RND of every size outperforms vanilla JTT. Selecting the size hyperparameter for NGRAM-RND using validation worst-group accuracy, like Liu et al. (2021) do for vanilla JTT, gives $n = 1$. At this size, JTT with NGRAM-RND outperforms vanilla JTT by 3% accuracy.

|  | Worst-group | Average |
|---|---|---|
| *Vanilla* JTT | 71.3% | 79.1% |
| PREM-MASK | 72.1% | 79.9% |
| 1-gram | 74.3% | 79.7% |
| 2-gram | 71.9% | 80.0% |
| 3-gram | 72.0% | 80.1% |
| 4-gram | 73.4% | 80.4% |
| ERM | 67.9% | − |

**Table 15:** ANLI (Nie et al., 2019) evaluations of models trained on MultiNLI. With a t-test to measure statistical significance, at the standard significance level of 0.05, we found that POE with NGRAM-RND gave a statistically significant improvement over the baseline on ANLI-R1 and ANLI-R2, while DFL gave a statistically significant improvement on ANLI-R1.

| Model | ANLI - R1 | ANLI - R2 | ANLI - R3 |
|---|---|---|---|
| ERM | $23.1 \pm 0.9$ | $28.2 \pm 0.8$ | $29.8 \pm 0.4$ |
| POE-known | $23.5 \pm 0.6$ | $27.8 \pm 0.8$ | $29.8 \pm 0.8$ |
| DFL-known | $23.7 \pm 1.3$ | $27.8 \pm 1.1$ | $30.4 \pm 0.9$ |
| POE - n3 | $24.8 \pm 1.1$ | $29.2 \pm 0.4$ | $30.4 \pm 1.2$ |
| DFL - n3 | $24.9 \pm 0.6$ | $29.0 \pm 1.2$ | $29.9 \pm 0.3$ |
| POE - PREM-MASK | $23.6 \pm 1.2$ | $27.3 \pm 0.8$ | $29.8 \pm 0.8$ |
| DFL - PREM-MASK | $22.3 \pm 0.7$ | $27.7 \pm 0.6$ | $29.3 \pm 1.1$ |

**Table 16:** Mean and standard deviation of CAD (Kaushik et al., 2019) test accuracy over 4 seeds. At the end, we also report the results of finetuning BERT on CAD training data from (Kaushik et al., 2019). When trained on MNLI, on average over the CAD subsets RH and RH, DFL and POE with semantic corruptions, DFL and POE with known-nuisances, and ERM perform on par (within one std.) or better than finetuning directly on the training CAD dataset. The improvement over finetuning directly on CAD may be due to the fact that the CAD dataset is much smaller than MNLI ( *7k* vs. *400k*).

| Method | RP | RH | Avg. on RP and RH |
|---|---|---|---|
| ERM on MNLI | $61.1 \pm 0.3$ | $76.5 \pm 0.4$ | $68.8 \pm 0.2$ |
| POE-known | $60.6 \pm 0.5$ | $77.0 \pm 1.1$ | $68.8 \pm 0.3$ |
| POE 3-gram | $60.8 \pm 0.5$ | $76.1 \pm 0.7$ | $68.4 \pm 0.2$ |
| POE PREM-MASK | $61.7 \pm 0.6$ | $75.6 \pm 1.0$ | $68.6 \pm 0.5$ |
| DFL-known | $60.6 \pm 0.8$ | $76.2 \pm 0.7$ | $68.4 \pm 0.4$ |
| DFL 3-gram | $58.4 \pm 1.8$ | $72.7 \pm 1.0$ | $65.5 \pm 1.4$ |
| DFL PREM-MASK | $62.4 \pm 0.7$ | $76.1 \pm 0.8$ | $69.3 \pm 0.6$ |
| ERM on CAD (from (Kaushik et al., 2019)) | 64.6 | 67.8 | 66.2 |

