# OpenReview forum: "Nuisances via Negativa: Adjusting for Spurious Correlations via Data Augmentation"
_TMLR — Accepted by TMLR_

### Review · Reviewer_bQk6 · 2024-01-13

**Summary Of Contributions:**

The contribution of this work is hard to assemble in one statement the way the paper is written. It would be very useful if the authors listed out short bullet points of the contribution.

**Audience:**

No

**Claims And Evidence:**

No

**Requested Changes:**

Improve the clarity of the contributions, possibly with short bullet points. Please state this directly and succinctly.

Because I am not clear on what the contributions of the paper are I don't believe it is ready for publication.

**Strengths And Weaknesses:**

The paper states "We develop an approach to use knowledge about the semantics via data augmentations." but this idea has been published previously as discussed by the authors (ie. NURD). Then the authors state "We demonstrate the value of semantic corruption by using it to power a variety of b-scams" so it is confusing what the contribution is. Is this a survey paper?

How is this work different from https://openreview.net/forum?id=12RoR2o32T which is cited by the authors as the basis for their experiments.

---

### Review · Reviewer_rdbj · 2024-02-08

**Summary Of Contributions:**

Previous work has developed biased-model-based spurious-correlation-avoiding methods (B-SCAMs) to avoid reliance on nuisances, i.e. features that would shift between the train and test domains. B-SCAMs require access to a biased model, a model that predicts the labels using the nuisances. The submission provides data corruption strategies that retain nuisances for training biased models to be used in B-SCAMs. The first type of corruption permutes batches in images and N-grams in text. The other types mask inputs in a certain region, certain frequency band, or certain range of brightness. The proposals can be used in conjunction with previous methods and these additions are shown to improve performance in the experiments. While the corruptions require some domain knowledge about nuisances, Theorem 1 shows that robustness to domain shift without any information about nuisances is not possible. Proposition 1 then shows that the overall procedure is effective if a model trained on the corrupted data is close to a model trained on nuisances.

**Audience:**

Yes

**Claims And Evidence:**

Yes

**Requested Changes:**

The comments on presentation and novelty are requested changes for acceptance. The comment on correctness is a suggestion.

**Strengths And Weaknesses:**

I do not have a strong background in this line of work so this is a low confidence review. The survey of previous work is at times too informal and it is hard to have a crisp idea of the background from the submission. I have clarified some points that could help with the presentation. The proposal is somewhat incremental and domain-specific but appropriate for TMLR and the motivation and the results are promising. The final decision would depend on the discussions with the other reviewers and the authors.

Presentation:

1. The description of previous work is too informal and scattered in my opinion. Ideally, one section would describe an example B-SCAM method (or a prototypical abstract B-SCAM method that would encompass multiple examples) step by step in clear algebraic form or perhaps pseudocode along with a sample application. Such section would clarify, for example, what a nuisance annotation is, how it is employed in the algorithms that require it, or why reweighting the loss would create robustness to nuisances. Right now the reader will have to move back and forth between sections or even different papers to build this background.
2. The definitions at the bottom of page 4 are claimed to break certain relationships. This part should elaborate further on why these distributions break these relationships.
3. The proposals are called a form of "data augmentation". Given that the corrupted data is used separately to train the biased model, I am not sure whether calling the method "augmentation" is appropriate. I leave the decision to the authors who are more familiar with the conventions in this line of research.

Correctness:

1. The effectiveness of the corruptions relies on breaking semantic information while keeping the nuisances. Evidence that this is in fact the reason behind the improved final performance would be helpful. The authors can perhaps use extra information in the dataset such as nuisance annotations to evaluate the obtained biased models and their reliance on semantics and nuisances. Another approach would be a qualitative analysis of some examples of success and failure to understand the reasons behind it. Such an analysis would be a great addition although not necessary for acceptance in my opinion.

Novelty:

1. The description of the result by Dormann et al below Proposition 1 should better clarify how Proposition 1 differs from it.

---

### Review · Reviewer_WqJw · 2024-03-17

**Summary Of Contributions:**

This work studies the OOD generalization by exploiting the relationship between nuisance and labels. Different from previous methods, the authors propose several methods to corrupt the semantics in the input, therefore, the model that can still predict the training labels is likely to rely on the nuisance. Then, they verify the effectiveness of the proposed method with different choices of corruption and datasets.

**Audience:**

Yes

**Claims And Evidence:**

Yes

**Requested Changes:**

1. It would be better if a summary of the contributions are provided in the paper.

2. To verify the effectiveness of the method in a more realistic setting, it would be better if some additional experiments
- with a nuisance correlation weaker than that of the invariance as discussed in [1,2,3] that show B-scams methods will fail,
- with multiple nuisance features [4],
- with different strengths of the corruption.

**References**

[1] Rich Feature Construction for the Optimization-Generalization Dilemma, ICML'22.

[2] Pareto Invariant Risk Minimization, ICLR'23.

[3] Understanding and Improving Feature Learning for Out-of-Distribution Generalization, NeurIPS'23.

[4] Spurious Feature Diversification Improves Out-of-distribution Generalization, arXiv'23.

**Strengths And Weaknesses:**

(+) The studied problem is important;

(+) The proposed method is new and interesting;

(+) The authors conduct comprehensive experiments to verify the effectiveness of the method;

(-) The proposed method relies on the assumption of the availability of knowledge about the nuisance;

---

### Decision · Action_Editor_aFDa · 2024-05-01

**Recommendation:** Accept as is

**Comment:**

The majority of reviewer recommend acceptance and the AE agrees. The proposed work will likely find an audience in TMLR readers and seems technically sound. The proposed abstractions of semantic corruptions as they relate to B-SCAM methods is an interesting framework and the empirical results demonstrate the proposed approaches are useful. The initial draft was insufficiently clear on the contributions of the paper and could have been mistaken as a survey paper from the introduction alone -- leading to some confusion. The most recent revision has responded to reviewer concerns about this and is clearer.

Reviewer bQk6 opposed acceptance due to unclear novelty over prior work by Puli et al. in ICLR 2022. As the authors explain the response, this manuscript can be viewed as an extension and generalization of the principles explored in Puli, 2022. The additional abstraction of semantic corruption powering biased-model-based spurious correlation avoiding methods is a useful framing and the resulting experiments are broader in terms of datasets and methodologies.

**Audience:**

The majority of reviewers agree there is an audience for this work and the AE agrees.

One aspect of having an audience is providing novelty -- not necessary in terms of technique, but rather in terms of providing useful information to readers that adds to (rather than simply reinforces) our collective understanding of a topic. Some results here could arguably be considered known (or predictable) by the community, spurring two reviewers to make note of this in the discussion. However, the AE believes sufficient useful information is provided in the submission to have an audience at TMLR.

**Claims And Evidence:**

The majority of reviewers agree the claims are supported by evidence. The dissenting reviewer does not provide sufficient reasoning to clarify what specific claims are unsupported. Considering the discussion and manuscript, the AE agrees that the primary claims are well-supported. The proposed semantic corruptions require limited domain knowledge and no annotation, but lead to models that avoid relying on nuisance variables comparably to methods that are nuisance-aware in some way.